# Discrete Adjoint Schrödinger Bridge Sampler

Wei Guo [1] [*]  Yuchen Zhu [1] [‡]  Xiaochen Du [2] [‡]  Juno Nam [2] [‡]  Yongxin Chen [1] [†]  Rafael Gómez-Bombarelli [2] [†]
Guan-Horng Liu [3] [†]  Molei Tao [1] [†]  Jaemoo Choi [1] [*]

## Abstract

Learning discrete neural samplers is challenging due to the lack of gradients and combinatorial complexity. While stochastic optimal control (SOC) and Schrödinger bridge (SB) provide principled solutions, efficient SOC solvers like adjoint matching (AM), which excel in continuous domains, remain unexplored for discrete spaces. We bridge this gap by revealing that the core mechanism of AM is *state-space agnostic*, and introduce **discrete ASBS**, a unified framework that extends AM and adjoint Schrödinger bridge sampler (ASBS) to discrete spaces. Theoretically, we analyze the optimality conditions of the discrete SB problem and its connection to SOC, identifying a necessary cyclic group structure on the state space to enable this extension. Empirically, discrete ASBS achieves competitive sample quality with significant advantages in training efficiency and scalability. Our code is available at https://github.com/AlexandreGUO2001/DASBS.

## 1. Introduction

Sampling from unnormalized distributions is a fundamental problem across computational statistics (Liu, 2008; Brooks et al., 2011), Bayesian inference (Gelman et al., 2013), and statistical mechanics (Landau & Binder, 2014). The goal is to draw samples from a target distribution $\nu$ on a state space $\mathcal{X}$, where $\nu(x) \propto \mathrm{e}^{-\beta E(x)}$ is specified through an energy function $E : \mathcal{X} \to \mathbb{R}$ and inverse temperature $\beta > 0$. In high-dimensional settings with complex energy landscapes, traditional methods such as Markov chain Monte Carlo (MCMC) suffer from slow mixing and poor scalability. Recent advances in **neural samplers** address these challenges by parameterizing sampling dynamics with deep neural networks, enabling efficient learning even without i.i.d. samples from the target. Crucially, these methods provide amortized inference, replacing costly MCMC iterations with rapid generation from a pretrained model.

For continuous state space $\mathcal{X} = \mathbb{R}^D$, neural samplers have achieved remarkable success through diverse methodologies (He et al., 2025; Sanokowski et al., 2025b; Blessing et al., 2026), including sequential Monte Carlo (Phillips et al., 2024), escorted transport (Vargas et al., 2024; Albergo & Vanden-Eijnden, 2025; Chen et al., 2025; Blessing et al., 2025b; Du et al., 2025), stochastic optimal control (SOC, Zhang & Chen (2022); Vargas et al. (2023); Richter & Berner (2024)), parallel tempering (Rissanen et al., 2025; Akhound-Sadegh et al., 2025; Zhang et al., 2026), etc. Among SOC approaches, **adjoint matching** (**AM**, Domingo-Enrich et al. (2025a); Havens et al. (2025)) has emerged as a powerful solver, enabling the framework of **adjoint Schrödinger bridge sampler** (**ASBS**, Liu et al. (2025a)), which offers fast convergence, scalability, and flexibility in reference dynamics. However, AM relies fundamentally on continuous calculus, making its extension to discrete state spaces highly nontrivial.

Motivated by these successes, recent work has explored discrete neural samplers based on continuous-time Markov chains (CTMCs), proposing similar methods based on escorted transport (Holderrieth et al., 2025; Ou et al., 2025b) and SOC formulations (Sanokowski et al., 2025a; Zhu et al., 2025a; Guo et al., 2026). Despite this progress, the Schrödinger Bridge (SB) problem – a distributionally constrained optimization framework central to optimal transport (Léonard, 2014; Chen et al., 2016b; 2021) and continuous diffusion models (Chen et al., 2021; De Bortoli et al., 2021; Chen et al., 2022; Liu et al., 2022; Shi et al., 2023; Theodoropoulos et al., 2026) – remains underdeveloped for discrete spaces (Ksenofontov & Korotin, 2025; Kim et al., 2025). The theoretical connection between discrete SB and SOC is not fully established, and no discrete SOC solver achieves the efficiency of continuous AM. While a recent work by So et al. (2026) attempted to adapt AM to masked discrete diffusion, their approach yields complex training objectives,

---

[*]Core contributors, equal contribution [‡]Co-second authors, equal contribution [†]Equal advising, alphabetical order [1]Georgia Institute of Technology [2]Massachusetts Institute of Technology [3]Meta Superintelligence Labs. Correspondence to: Wei Guo <wei.guo@gatech.edu>, Jaemoo Choi <jchoi843@gatech.edu>.

*Proceedings of the 43rd International Conference on Machine Learning*, Seoul, South Korea. PMLR 306, 2026. Copyright 2026 by the author(s).

*Table 1.* Conceptual comparison between denoising matching and adjoint matching in continuous and discrete state spaces. Denoising matching holds for general reference dynamics while adjoint matching requires boxed(additive) noise.

| | Continuous state space $\mathcal{X} = \mathbb{R}^D$ | Discrete state space $\mathcal{X} = \mathbb{Z}_N^D$ |
|---|---|---|
| Ref. dyn. $p^r$ | $\mathrm{d}X_t = b_t(X_t)\mathrm{d}t + \sigma_t\mathrm{d}W_t,\ X_0 \sim \mu$ | CTMC $(X_t)$ with transition rate $(r_t),\ X_0 \sim \mu$ |
| Ctrl. dyn. $p^u$ | $\mathrm{d}X_t = (b_t + \sigma_t u_t)(X_t)\mathrm{d}t + \sigma_t\mathrm{d}W_t,\ X_0 \sim \mu$ | CTMC $(X_t)$ with transition rate $(u_t),\ X_0 \sim \mu$ |
| SB Prob. | $\min\limits_{u \text{ s.t. } X_1 \sim \nu} \mathbb{E}_{X \sim p^u}\Big[\int\limits_0^1 \frac{1}{2}\|u_t(X_t)\|^2\mathrm{d}t\Big]$ | $\min\limits_{u \text{ s.t. } X_1 \sim \nu} \mathbb{E}_{X \sim p^u}\Big[\int\limits_0^1 \sum\limits_{y \neq X_t} \big(u_t \log \frac{u_t}{r_t} + r_t - u_t\big)(y, X_t)\mathrm{d}t\Big]$ |
| SOC Prob. | $\min\limits_{u} \mathbb{E}_{X \sim p^u}\Big[\int\limits_0^1 \frac{1}{2}\|u_t(X_t)\|^2\mathrm{d}t + \log \frac{\widehat{\varphi}_1}{\nu}(X_1)\Big]$ | $\min\limits_{u} \mathbb{E}_{X \sim p^u}\Big[\int\limits_0^1 \sum\limits_{y \neq X_t} \big(u_t \log \frac{u_t}{r_t} + r_t - u_t\big)(y, X_t)\mathrm{d}t + \log \frac{\widehat{\varphi}_1}{\nu}(X_1)\Big]$ |
| Opt. Ctrl. | $u_t^\star(x) = \sigma_t \nabla \log \varphi_t(x)$ | $u_t^\star(y, x) = r_t(y, x)\frac{\varphi_t(y)}{\varphi_t(x)}$ |
| Corrector | $\nabla \log \widehat{\varphi}_1(x)$ | $\frac{\widehat{\varphi}_1(y)}{\widehat{\varphi}_1(x)}$ |
| Example of boxed(Additive) Noise | $b_t \equiv 0 \Rightarrow \boxed{p_{1\mid t}^r(x_1\mid x) = q_t(x_1 - x)}$ 
 $q_t(\varepsilon) = \mathcal{N}(\varepsilon; 0, \widetilde{\sigma}_t^2 I)$ | $r_t(y, x) = \frac{\gamma_t}{N}\mathbb{1}_{d_{\mathrm{H}}(x,y)=1} \Rightarrow \boxed{p_{1\mid t}^r(x_1\mid x) = q_t(x_1 - x)}$ 
 $q_t(\varepsilon) = A(t, 1)^{d_{\mathrm{H}}(\varepsilon, 0)}B(t, 1)^{D - d_{\mathrm{H}}(\varepsilon, 0)}$ |
| **Denoising Matching** | $\nabla \log \varphi_t(x) = \mathbb{E}_{p_{1\mid t}^\star(x_1\mid x)}\nabla_x \log p_{1\mid t}^r(x_1\mid x)$ (1) 
 $\nabla \log \widehat{\varphi}_1(x) = \mathbb{E}_{p_{0\mid 1}^\star(x_0\mid x)}\nabla_x \log p_{1\mid 0}^r(x\mid x_0)$ (2) | $\frac{\varphi_t(y)}{\varphi_t(x)} = \mathbb{E}_{p_{1\mid t}^\star(x_1\mid x)}\frac{p_{1\mid t}^r(x_1\mid y)}{p_{1\mid t}^r(x_1\mid x)}$ 
 $\frac{\widehat{\varphi}_1(y)}{\widehat{\varphi}_1(x)} = \mathbb{E}_{p_{0\mid 1}^\star(x_0\mid x)}\frac{p_{1\mid 0}^r(y\mid x_0)}{p_{1\mid 0}^r(x\mid x_0)}$ |
| **Adjoint Matching** | $\nabla \log \varphi_t(x) = \mathbb{E}_{p_{1\mid t}^\star(x_1\mid x)}\nabla \log \varphi_1(x_1)$ (3) 
 $\nabla \log \widehat{\varphi}_1(x) = \mathbb{E}_{p_{0\mid 1}^\star(x_0\mid x)}\nabla \log \widehat{\varphi}_0(x_0)$ (4) | $\frac{\varphi_t(y)}{\varphi_t(x)} = \mathbb{E}_{p_{1\mid t}^\star(x_1\mid x)}\frac{\varphi_1(y + x_1 - x)}{\varphi_1(x_1)}$ 
 $\frac{\widehat{\varphi}_1(y)}{\widehat{\varphi}_1(x)} = \mathbb{E}_{p_{0\mid 1}^\star(x_0\mid x)}\frac{\widehat{\varphi}_0(y + x_0 - x)}{\widehat{\varphi}_0(x_0)}$ |

leaving a principled and efficient discrete AM framework an open challenge.

In this work, we develop a unified theoretical framework for SB and SOC on discrete state spaces, introducing the **discrete adjoint Schrödinger bridge sampler (DASBS)**. We derive optimality conditions for the discrete SB problem that structurally mirror the continuous setting, and extend AM to discrete domains through novel controller and corrector learning objectives. Our approach reveals that a group structure on the state space and uniform reference dynamics are essential requirements that parallel the additive noise assumption in continuous AM. Our **key contributions** are:

**I. Unified Discrete Framework**: We formalize discrete neural sampling as an SB problem for CTMCs and establish its equivalent SOC formulation.

**II. Generalizing Adjoint Matching**: We identify state-space agnostic principles underlying AM, providing conceptual clarity for extensions beyond continuous spaces.

**III. Discrete ASBS**: We introduce discrete ASBS, a principled algorithm that alternates between adjoint and corrector matching, derived from variational characterizations of the learning objectives.

**IV. Empirical Validation**: We validate DASBS on high-dimensional discrete benchmarks, demonstrating competitive performance and efficiency.

## 2. Preliminaries

**Diffusion Samplers on Continuous State Space** $\mathcal{X} = \mathbb{R}^D$ MCMC sampling based on equilibrium dynamics, e.g., Langevin Monte Carlo, is typically slow to mix when the target distribution is complex. Recent progress in non-equilibrium measure transport for generative modeling such as diffusion models (Song et al., 2021) has motivated sampling methods based on controlled stochastic differential equations (SDEs), commonly referred to as diffusion neural samplers. The sampling dynamics are described by an SDE:

$$p^u:\ \mathrm{d}X_t = (b_t + \sigma_t u_t)(X_t)\mathrm{d}t + \sigma_t\mathrm{d}W_t,\ X_0 \sim \mu, \quad (5)$$

where $b : [0, 1] \times \mathcal{X} \to \mathcal{X}$ is the base drift, $\sigma : [0, 1] \to \mathbb{R}_{>0}$ is the noise schedule, and $\mu$ is the initial source distribution. Given $(b_., \sigma_., \mu)$, the goal is to learn a parameterized control $u : [0, 1] \times \mathcal{X} \to \mathcal{X}$ such that the marginal distribution of $X_1$ matches the target distribution $\nu$. We use $p^u$ to denote the **path measure** induced by this SDE with control $u$. Formally, $p^u(X_{[0,1]})$ can be viewed as the limit of the joint distribution of $(X_{t_0}, X_{t_1}, ..., X_{t_K})$ as the partition $0 = t_0 < t_1 < ... < t_K = 1$ becomes finer and finer, and a rigorous definition is through the Radon-Nikodým derivative (RND).

**Schrödinger Bridge Problem on Continuous State Spaces** One way to formulate the learning of diffusion samplers is through a distributionally constrained optimal transport problem known as the **Schrödinger bridge (SB)** problem (Léonard, 2014; Chen et al., 2016b):

$$\min_{u \text{ s.t. } X_1 \sim \nu}\Big\{\mathrm{KL}(p^u\|p^r) = \mathbb{E}_{X \sim p^u}\int_0^1 \frac{1}{2}\|u_t(X_t)\|^2\mathrm{d}t\Big\}, \quad (6)$$

where $p^u$ is the controlled path measure (5) and $p^r$ is the path measure of a *reference dynamics* with zero control ($u \equiv 0$). The optimal control can be characterized by

$$u_t^\star(x) = \sigma_t \nabla \log \varphi_t(x), \tag{7}$$

where the **SB potentials** $(\varphi_t, \widehat{\varphi}_t)$ are defined (up to multiplicative constants) through time integrations with respect to the reference transition kernel $p_{t|s}^r(y|x)$:

$$\varphi_t(x) = \int p_{1|t}^r(y|x)\varphi_1(y)\mathrm{d}y, \quad \varphi_0(x)\widehat{\varphi}_0(x) = \mu(x);$$
$$\widehat{\varphi}_t(x) = \int p_{t|0}^r(x|y)\widehat{\varphi}_0(y)\mathrm{d}y, \quad \varphi_1(x)\widehat{\varphi}_1(x) = \nu(x).$$

**Stochastic Optimal Control Characteristics of SB** An interesting connection between the SB formulation and **stochastic optimal control (SOC)** is revealed by the characterization of the optimal control $u^\star$. As shown in Dai Pra (1991); Chen et al. (2016b); Liu et al. (2025a), $u^\star$ (7) is also the solution to the following SOC problem:

$$\min_u \mathbb{E}_{X \sim p^u}\left[\int_0^1 \frac{1}{2}\|u_t(X_t)\|^2\mathrm{d}t + \log\frac{\widehat{\varphi}_1(X_1)}{\nu(X_1)}\right]. \tag{8}$$

(8) highlights that SB (6) admits an SOC interpretation, where the terminal marginal constraint is encoded through the terminal cost $\log\frac{\widehat{\varphi}_1}{\nu}$. This perspective provides a useful bridge between SB theory and SOC formulations.

**Adjoint Schrödinger Bridge Sampler (ASBS, Liu et al. (2025a))** When $b_t \equiv 0$, the reference dynamics reduces to Brownian motion. In this setting, the reference transition kernel is **additive** in the sense that sampling from $p_{1|t}^r(\cdot|x)$ can be achieved by adding a noise $\epsilon \sim q_t$ onto $x$, i.e.,

$$p_{1|t}^r(y|x) = q_t(y - x), \quad q_t = \mathcal{N}(0, \widetilde{\sigma}_t^2 I), \tag{9}$$

where $\widetilde{\sigma}_t^2 = \int_t^1 \sigma_s^2 \mathrm{d}s$. Importantly, this additive property is central in simplifying the associated SOC problem (8); it yields tractable conditional distributions $p_{1|t}^r$ and enables explicit expressions for the quantities appearing in the optimality conditions (Havens et al., 2025). Exploiting this structure, Liu et al. (2025a) derived the **adjoint matching (AM)** identities (3) and (4), as well as the corresponding **denoising matching (DM)** identities (1) and (2), and proposed an alternating procedure based on (2) and (3) to learn the controller and the corrector, which converges under suitable regularity conditions. See the left part of Tab. 1 for details.

**Connection to the Memoryless Case (Adjoint Sampling)** An earlier work, adjoint sampling (Havens et al., 2025), considered a special case in which the reference path measure $p^r$ is **memoryless**, i.e. $p_{0,1}^r(x, y) = p_0^r(x)p_1^r(y)$, under which one can show that $\widehat{\varphi}_1 \propto p_1^r$. As a consequence, the corrector $\nabla \log \widehat{\varphi}_1$ is known and the learning problem

reduces to a single regression objective for the controller. From this perspective, ASBS is a generalization of adjoint sampling that relaxes the memoryless assumption, allowing for nontrivial boundary coupling. It is also observed in Liu et al. (2025a) that non-memoryless reference dynamics enable reduced noise levels compared with memoryless ones, thus offering improved performance.

## 3. SB and SOC Theory for CTMC

In this section, we introduce the SB problem for continuous-time Markov chains (CTMCs), serving as a discrete analog of (6). We then extend the corresponding SB and SOC theory from continuous state spaces to discrete CTMCs (Léonard, 2014). In particular, we derive optimality conditions for the optimal transition rate $u^\star$, analogous to (7), and formulate an associated SOC problem in the spirit of (8).

### 3.1. Problem Setting

**Continuous-time Markov chain** Throughout this paper, we will consider the discrete state space $\mathcal{X}$ as the set of length-$D$ sequences with $N$ possible states $[N] := \{1, 2, ..., N\}$, i.e., $\mathcal{X} = [N]^D$. A **continuous-time Markov chain (CTMC)** $(X_t)_{t \in [0,1]}$ is a stochastic process taking values in $\mathcal{X}$ and characterized by its **transition rate** $r = (r_t(y, x))_{t \in [0,1]}^{x,y \in \mathcal{X}}$, defined by

$$r_t(y, x) = \lim_{h \to 0} \frac{\Pr(X_{t+h} = y | X_t = x) - 1_{y=x}}{h},$$

where $1_A \in \{0, 1\}$ is the indicator of a statement $A$.

**SB Problem on Discrete State Spaces** Following Sec. 2, we address the sampling problem by formulating an SB problem for CTMCs, whose optimal path measure $p^\star$ satisfies the boundary marginal constraints $p_0^\star = \mu$ and $p_1^\star = \nu$. Let $r_t(y, x)$ and $u_t(y, x)$ denote the reference and controlled transition rates, and write $p^r$ and $p^u$ for their induced **path measures**. We assume a common initial distribution $p_0^r = p_0^u = \mu$. Consider the following SB problem for CTMCs:

$$\min_{u \text{ s.t. } p_1^u = \nu}\left\{\mathrm{KL}(p^u\|p^r) = \right.$$
$$\left. \mathbb{E}_{X \sim p^u}\int_0^1\sum_{y \neq X_t}\left(u_t\log\frac{u_t}{r_t} + r_t - u_t\right)(y, X_t)\mathrm{d}t\right\}. \tag{SB}$$

Next, we will make explicit the connection between this SB problem and a corresponding SOC formulation, in direct analogy with (8).

### 3.2. SB and SOC Theory for CTMC

**Characterization of the Optimal Transition Rate** We now characterize the optimal transition rate solving (SB).

The following result provides an explicit description of the optimal transition rate in terms of a pair of time-dependent potentials, which play a role analogous to the SB potentials in the continuous-state setting. See Sec. C.2 for the proof.

**Theorem 1.** *The optimal transition rate $u^\star$ for (SB) can be expressed as*

$$u_t^\star(y,x) = \frac{\varphi_t(y)}{\varphi_t(x)} r_t(y,x), \ \forall y \neq x, \qquad (10)$$

*where the **SB potentials** $(\varphi_t, \widehat{\varphi}_t)$ satisfy*

$$\forall 0 \leq s < t \leq 1 : \begin{cases} \varphi_s(x) = \sum_y p_{t|s}^r(y|x)\varphi_t(y), & (11) \\ \widehat{\varphi}_t(x) = \sum_y p_{t|s}^r(x|y)\widehat{\varphi}_s(y), & (12) \end{cases}$$

*and the optimal path measure $p^\star$ satisfies*

$$p_t^\star = \varphi_t \widehat{\varphi}_t, \ \frac{p_{t|s}^\star(x|y)}{p_{t|s}^r(x|y)} = \frac{\varphi_t(x)}{\varphi_s(y)}, \ \frac{p_{s|t}^\star(y|x)}{p_{t|s}^r(x|y)} = \frac{\widehat{\varphi}_s(y)}{\widehat{\varphi}_t(x)}. \quad (13)$$

(11) and (12) are a forward-backward representation with respect to the reference transition kernel, and the boundary marginal constraints are encoded through the coupling conditions at boundary times: $\varphi_0\widehat{\varphi}_0 = \mu$, $\varphi_1\widehat{\varphi}_1 = \nu$.

**SOC Problem on Discrete State Spaces**  The SOC problem for CTMCs with terminal cost $g : \mathcal{X} \to \mathbb{R}$ is

$$\min_{u \text{ s.t. } p_0^u = \mu} \left\{ \mathrm{KL}(p^u \| p^r) + \mathbb{E}_{X \sim p^u} g(X_1) \right\} \qquad (\text{SOC})$$

**SOC Characteristics of SB**  While the optimality conditions in (10) provide an explicit characterization of $u^\star$, they are challenging to solve in practice. The main difficulties are twofold: the coupled boundary constraints at times $t = 0$ and $t = 1$, and the need to evaluate expectations with respect to the reference transition kernels. The SOC formulation circumvents these issues by avoiding the direct solution of coupled equations. The following theorem establishes an SOC reinterpretation of the SB problem:

**Theorem 2.** *The optimal transition rate $u_t^\star$ (10) for (SB) solves (SOC) with terminal cost $g \leftarrow \log \frac{\widehat{\varphi}_1}{\nu}$.*

*Sketch of proof.* The optimal path measures of (SB) and (SOC) can be respectively written as

$$\frac{p_{SB}^\star(X_{[0,1]})}{p^r(X_{[0,1]})} = \frac{\widehat{\varphi}_0(X_0)}{\mu(X_0)} \frac{\nu(X_1)}{\widehat{\varphi}_1(X_1)}, \qquad (14)$$

$$\frac{p_{SOC}^\star(X_{[0,1]})}{p^r(X_{[0,1]})} = \frac{e^{-g(X_1)}}{Z(X_0)}, \ Z(x) = \mathbb{E}_{p_{1|0}^r(y|x)} e^{-g(y)}. \quad (15)$$

See Sec. C.4 for the full proof. Thm. 2 shows that (SB) admits an equivalent SOC formulation in which the terminal marginal constraint $p_1^u = \nu$ is incorporated through the terminal cost $g = \log \frac{\widehat{\varphi}_1}{\nu}$. This SOC perspective will be instrumental for developing tractable learning objectives in the discrete setting.

**Connection to Memoryless Reference Dynamics**  If we further assume $p^r$ is **memoryless**, then (11) implies $\varphi_0(x) = \mathbb{E}_{p_1^r} e^{-g} = \text{const}$, and (12) implies $\widehat{\varphi}_1(x) = p_1^r(x) \sum_y \widehat{\varphi}_0(y) \propto p_1^r(x)$. Therefore, $g = \log \frac{p_1^r}{\nu} + \text{const}$.

**Relation to Continuous State Spaces**  Finally, we note that the SOC and SB theory developed above for discrete state spaces closely parallels its continuous counterpart introduced in Sec. 2. A detailed comparison is provided in the upper part of Tab. 1.

# 4. Discrete Adjoint Schrödinger Bridge Sampler (DASBS)

In this section, we introduce a principled theory and algorithm for learning the optimal transition rates in the discrete SB problem. Building on the SOC and SB formulation for CTMCs developed in the previous section, we propose a discrete analogue of ASBS (Liu et al., 2025a) by developing discrete versions of adjoint matching and denoising matching for controller and corrector.

A key challenge in the discrete setting is the absence of additive noise structure (9) that plays a central role in developing an AM framework. To address this issue, we will adopt a *cyclic group structure* on the discrete state space, which allows discrete transitions to be interpreted in an additive form. Under this perspective, a uniform reference transition rate emerges as a natural choice for inducing tractable and symmetric transition kernels.

**Choice of the Reference Path Measure**  (10) implies that it suffices to learn the ratio of $\varphi_t$ at all pairs of $x, y \in \mathcal{X}$ such that $r_t(y,x) > 0$. We follow the typical strategy in discrete diffusion models (Campbell et al., 2022; Lou et al., 2024; Schiff et al., 2025) to restrict the transition to pairs of $x, y$ with Hamming distance $d_H(x,y) = 1$, i.e., $x$ and $y$ differ at exactly one entry. Throughout this paper, we consider the reference path measure $p^r$ starting from an arbitrary tractable initial distribution $p_0^r = \mu$ and induced by the following **uniform transition rate** $r$ that keeps the uniform distribution on $\mathcal{X}$ ($p_{\text{unif}} = \frac{1}{N^D}$) invariant:

$$r_t(y,x) = \begin{cases} \frac{\gamma_t}{N}, & \text{if } d_H(y,x) = 1, \\ -\gamma_t D \left(1 - \frac{1}{N}\right), & \text{if } y = x, \\ 0, & \text{if otherwise,} \end{cases} \qquad (16)$$

where $\gamma_\cdot : [0,1] \to \mathbb{R}_+$ is a noise schedule. We remark that for two given time steps $0 \le s < t \le 1$ and states $x, y \in \mathcal{X}$, $p^r_{t|s}(y|x)$ can be written in a closed form (Prop. 3), and so is $p^r_{t|0,1}(x|x_0, x_1)$ (Prop. 4). See Sec. C.5 for details.

Thus, it suffices to learn $\frac{\varphi_t(y)}{\varphi_t(x)}$ for all $d_{\mathrm{H}}(x, y) = 1$. Define the **controller** matrix $\Phi^\star_t(x) \in \mathbb{R}^{D \times N}$ whose $(d, n)$-th element is $\Phi^\star_t(x)_{d,n} = \frac{\varphi_t(x^{d \leftarrow n})}{\varphi_t(x)}$, where $x^{d \leftarrow n}$ denotes the vector obtained by replacing the $d$-th entry of $x$ with $n$.

**Cyclic Group Structure for the State Space**  To extend AM (3) to discrete space, we treat the state space $\mathcal{X} = [N]^D$ as $\mathbb{Z}_N^D$, where $\mathbb{Z}_N$ is the cyclic group of integers modulo $N$. Thus, the sum and difference of any two elements in $\mathcal{X}$ are still in $\mathcal{X}$. The key benefit of doing so is that the transition kernel $p^r_{1|t}(\cdot|x)$ is again **additive**:

$$\exists q_t \text{ (see Prop. 3)} \quad \text{s.t.} \quad p^r_{1|t}(y|x) = q_t(y - x). \quad (17)$$

We provide further explanation and justification for this cyclic group structure in Sec. B.

**Controller Adjoint Matching**  With additive noise, we can thus consider a target score matching (De Bortoli et al., 2024; Zhang et al., 2025) objective like in the continuous AM (Domingo-Enrich et al., 2025a; Havens et al., 2025). Applying (17), we can refactor (11) as follows:

$$
\begin{aligned}
\varphi_t(y) &\underset{(11)}{=} \sum_{x_1 \in \mathbb{Z}_N^D} p^r_{1|t}(x_1|y) \varphi_1(x_1) \\
&\underset{(17)}{=} \sum_{x_1 \in \mathbb{Z}_N^D} p^r_{1|t}(x_1 + \Delta | y + \Delta) \varphi_1(x_1) \\
&\underset{x_1' \leftarrow x_1 + \Delta}{=} \sum_{x_1' \in \mathbb{Z}_N^D} p^r_{1|t}(x_1' | y + \Delta) \varphi_1(x_1' - \Delta) \\
&\underset{\Delta \leftarrow x - y}{=} \sum_{x_1' \in \mathbb{Z}_N^D} p^r_{1|t}(x_1' | x) \varphi_1(x_1' - x + y).
\end{aligned} \quad (18)
$$

$$
\begin{aligned}
\implies \frac{\varphi_t(y)}{\varphi_t(x)} &\underset{(18)}{=} \mathbb{E}_{p^r_{1|t}(x_1|x)} \frac{\varphi_1(x_1 + y - x)}{\varphi_t(x)} \\
&\underset{(13)}{=} \mathbb{E}_{p^\star_{1|t}(x_1|x)} \frac{\varphi_1(x_1 + y - x)}{\varphi_1(x_1)}.
\end{aligned} \quad (19)
$$

When $y \leftarrow x^{d \leftarrow n}$, since $x^{d \leftarrow n} - x$ has at most one nonzero coordinate, $x_1$ and $x_1 + x^{d \leftarrow n} - x = x_1^{d \leftarrow x_1^d + n - x^d}$ differ in at most one entry. Let $D_f(a\|b) = f(a) - f(b) - (a - b)f'(b) \ge 0$ be the Bregman divergence induced by a strictly convex and differentiable function $f$, and note that $\mathbb{E}\xi = \arg\min_{c \in \mathbb{R}} \mathbb{E} D_f(\xi\|c)$ for any random variable $\xi$ (Lou et al., 2024). We thus have the following variational characterization of the controller $\Phi^\star$:

$$
\Phi^\star = \arg\min_\Phi \mathbb{E}_t \mathbb{E}_{p^\star_{t,1}(x, x_1)}
$$

$$
\sum_{d=1}^{D} \sum_{n \neq x^d} D_f \left( \frac{\varphi_1(x_1^{d \leftarrow x_1^d + n - x^d})}{\varphi_1(x_1)} \middle\| \Phi_t(x)_{d,n} \right), \quad (20)
$$

where $t$ is a random variable in $(0, 1)$ and one can sample $p^\star_{t,1}(x, x_1)$ by

$$
\begin{aligned}
p^\star_{0,t,1}(x_0, x, x_1) &= p^\star_{0,1}(x_0, x_1) p^\star_{t|0,1}(x|x_0, x_1) \\
&\underset{(26)}{=} p^\star_{0,1}(x_0, x_1) p^r_{t|0,1}(x|x_0, x_1).
\end{aligned}
$$

We can further rewrite the ratio in (20) as follows: [1]

$$
\frac{\varphi_1(x_1^{d \leftarrow \triangle})}{\varphi_1(x_1)} \underset{(13)}{=} \frac{\nu(x_1^{d \leftarrow \triangle})}{\nu(x_1)} \middle/ \underbrace{\frac{\widehat{\varphi}_1(x_1^{d \leftarrow \triangle})}{\widehat{\varphi}_1(x_1)}}_{= \widehat{\Phi}^\star(x_1)_{d,\triangle}}. \quad (21)
$$

Notably, here, we require the **discrete score** $x \mapsto \left( \frac{\nu(x^{d \leftarrow n})}{\nu(x)} \right)_{d,n}$ of the target distribution, which is a *first-order* oracle similar to the score of the target distribution in continuous AM.

**Corrector Adjoint Matching**  From (20) and (21), to learn the controller $\Phi^\star_t$, we require estimating $\frac{\widehat{\varphi}_1(y)}{\widehat{\varphi}_1(x)}$ for all $d_{\mathrm{H}}(x, y) = 1$. Let the **corrector** matrix $\widehat{\Phi}^\star(x) \in \mathbb{R}^{D \times N}$ be defined by its entries $\widehat{\Phi}^\star(x)_{d,n} = \frac{\widehat{\varphi}_1(x^{d \leftarrow n})}{\widehat{\varphi}_1(x)}$. Following the spirit of (18), we can derive the following identity and variational characterization of $\widehat{\Phi}^\star$ (see Sec. C.7 for proof):

$$
\frac{\widehat{\varphi}_1(z)}{\widehat{\varphi}_1(y)} = \sum_x p^\star_{t|1}(x|y) \frac{\widehat{\varphi}_t(x - y + z)}{\widehat{\varphi}_t(x)}, \; \forall t \in [0, 1), \quad (22)
$$

$$
\implies \widehat{\Phi}^\star = \arg\min_{\widehat{\Phi}} \mathbb{E}_{p^\star_{t,1}(x, x_1)}
$$

$$
\sum_{d=1}^{D} \sum_{n \neq x^d} D_f \left( \frac{\widehat{\varphi}_t(x^{d \leftarrow x^d + n - x_1^d})}{\widehat{\varphi}_t(x)} \middle\| \widehat{\Phi}(x_1)_{d,n} \right). \quad (23)
$$

However, if $t \neq 0$, leveraging the relation (13) requires knowing the intractable $p^\star_t$; otherwise, as $\mu$ is known, assuming it is *fully supported* on $\mathcal{X}$, we have[1]

$$
\frac{\widehat{\varphi}_0(x^{d \leftarrow \square})}{\widehat{\varphi}_0(x)} \underset{(13)}{=} \frac{\mu(x^{d \leftarrow \square})}{\mu(x)} \middle/ \underbrace{\frac{\varphi_0(x^{d \leftarrow \square})}{\varphi_0(x)}}_{= \Phi^\star_0(x)_{d,\square}}. \quad (24)
$$

**Corrector Denoising Matching**  A limitation of (22) and (24) is that they require the explicit density of $\mu$ to be

---

[1] For conciseness, $\triangle := x_1^d + n - x^d$ and $\square := x^d + n - x_1^d$.

positive everywhere. To circumvent this, one can leverage the following identity:

$$\frac{\widehat{\varphi}_1(z)}{\widehat{\varphi}_1(y)} \underset{(12)}{=} \sum_x p_{1|t}^r(z|x)\frac{\widehat{\varphi}_t(x)}{\widehat{\varphi}_1(y)} \underset{(13)}{=} \sum_x \frac{p_{1|t}^r(z|x)}{p_{1|t}^r(y|x)}p_{t|1}^\star(x|y).$$
(25)

Notably, while (22) relies on the additive noise (17), (25) holds under general $p_{1|t}^r$. Thus, one can obtain a similar variational characterization of the corrector $\widehat{\Phi}^\star$ by replacing the regression target (i.e., the ratio) in (23) with $\frac{p_{1|t}^r(x_1^{d\leftarrow n}|x)}{p_{1|t}^r(x_1|x)}$. As this is related to the discrete score of the transition kernel $p_{1|t}^r(\cdot|x)$, we borrow the terminology in the continuous domain and call it **denoising matching (DM)**.

**Alternating Update**   We can thus arrive at the core algorithm of **discrete ASBS**, following the practice in ASBS to learn $\Phi \approx \Phi^\star$ and $\widehat{\Phi} \approx \widehat{\Phi}^\star$ with an alternating update. We initialize $\widehat{\Phi}^{(0)}$ to be all-one following the practice in Liu et al. (2025a), and for epoch $k = 1, 2, ...$, we solve the following two problems sequentially:

$$\Phi^{(k)} := \underset{\Phi}{\arg\min}\ \mathbb{E}_t w_t \mathbb{E}_{\substack{p_{0,1}^{\mathrm{sg}(r\Phi)}(x_0,x_1)\\ p_{t|0,1}^r(x|x_0,x_1)}} \quad \text{(ctrl-AM)}$$
$$\sum_{d=1}^{D}\sum_{n\neq x^d} D_f\left(\frac{\varphi_1(x_1^{d\leftarrow x_1^d+n-x^d})}{\varphi_1(x_1)}\middle\|\Phi_t(x)_{d,n}\right),$$

$$\widehat{\Phi}^{(k)} := \underset{\widehat{\Phi}}{\arg\min}\ \mathbb{E}_{p_{0,1}^{\mathrm{sg}(r\Phi^{(k)})}(x_0,x_1)} \quad \text{(corr-AM)}$$
$$\sum_{d=1}^{D}\sum_{n\neq x_1^d} D_f\left(\frac{\widehat{\varphi}_0(x_0^{d\leftarrow x_0^d+n-x_1^d})}{\widehat{\varphi}_0(x_0)}\middle\|\widehat{\Phi}(x_1)_{d,n}\right),$$

$$\text{or } \widehat{\Phi}^{(k)} := \underset{\widehat{\Phi}}{\arg\min}\ \mathbb{E}_t w_t \mathbb{E}_{\substack{p_{0,1}^{\mathrm{sg}(r\Phi^{(k)})}(x_0,x_1)\\ p_{t|0,1}^r(x|x_0,x_1)}} \quad \text{(corr-DM)}$$
$$\sum_{d=1}^{D}\sum_{n\neq x_1^d} D_f\left(\frac{p_{1|t}^r(x_1^{d\leftarrow n}|x)}{p_{1|t}^r(x_1|x)}\middle\|\widehat{\Phi}(x_1)_{d,n}\right).$$

The stop gradient operator $\mathrm{sg}(\cdot)$ applied onto a model means not tracking the gradient when querying the model. In (ctrl-AM) and (corr-DM), $t \sim \mathrm{Unif}(0,1)$, $w_. : [0,1] \rightarrow \mathbb{R}_+$ is a time weight function. For the two AM losses, the ratio in (ctrl-AM) is computed via (21) by replacing $\widehat{\Phi}^\star$ with the current $\mathrm{sg}(\widehat{\Phi}^{(k-1)})$, and the ratio in (corr-AM) is computed via (24) by replacing $\Phi^\star$ with the current $\mathrm{sg}(\Phi^{(k)})$. In all three losses, $p^{\mathrm{sg}(r\Phi)}$ means sampling from the CTMC with transition rate $u_t(x^{d\leftarrow n}, x) = r_t(x^{d\leftarrow n}, x)\,\mathrm{sg}(\Phi_t(x)_{d,n})$, $n \neq x^d$, which replaces $p_{0,1}^\star$ in the expectation with the detached non-optimal path measure. The theoretical justification of the validity of this replacement will be discussed in Thm. 3, and in principle, an optional **trajectory importance**

**reweighting** using the RND $p^\star(x_{[0,1]})/p^{\mathrm{sg}(r\Phi)}(x_{[0,1]})$ can be incorporated (Sec. C.9). In practice, we do not rely on this reweighting due to discretization and approximation errors of the estimated RND. Importantly, we will show in Thm. 3 that the unweighted alternating updates still converge to $p^\star$.

**Initialization**   Following ASBS (Liu et al., 2025a), one can initialize either the controller or the corrector to be non-informative (i.e., output all ones). In Prop. 8, we prove that under the reference transition rate (16) and uniform initialization $\mu = p_{\mathrm{unif}}$, these two approaches are equivalent, and hence we always start with the all-one corrector.

**Inference**   We use the **$\tau$-leaping** method (Gillespie, 2001; Campbell et al., 2022; Lou et al., 2024) to sample each dimension's transition independently. Since $u_t(x^{d\leftarrow n}, x) = \frac{\gamma_t}{N}\Phi_t(x)_{d,n}$, $n \neq x^d$, this means we fix $\Phi_\tau(X_\tau)_{d,n}$ on the interval $\tau \in [t, t+h]$ for a small step size $h > 0$, and assume each dimension evolves independently. We defer the details to Sec. C.6.

The full algorithm for training DASBS is detailed in Alg. 1.

## 5. Additional Theory and Insights of DASBS

In this section, we provide further theory and insights into the DASBS framework.

**AM v.s. DM for Corrector**   Recall that our derivation of the AM losses (ctrl-AM) and (corr-AM) leverages the relations (19) and (22), which rely explicitly on the additiveness of the reference transition kernel $p_{1|t}^r(\cdot|x)$ (17). In contrast, the DM loss (corr-DM) does not rely on this condition.

**AM v.s. DM for Controller**   A parallel relationship holds for the *controller*, which yields a different way of the alternating update by replacing the ratio in (ctrl-AM) with $\frac{p_{1|t}^r(x_1|x^{d\leftarrow n})}{p_{1|t}^r(x_1|x)}$ (see Sec. C.8 and (ctrl-DM) for details). However, though theoretically grounded, this formulation may suffer from a weak *mutual supervisory signal* during alternating updates: we rely on the trained $\widehat{\Phi}^{(k-1)}$ to supervise the training of $\Phi^{(k)}$, but such supervisory information only comes in from the *implicit boundary relation* $\Phi_1(x)_{d,n}\widehat{\Phi}(x)_{d,n} = \frac{\nu(x^{d\leftarrow n})}{\nu(x)}$, which is not explicitly enforced in the loss. Even when the reference dynamics is *memoryless* and there is no need to learn the corrector, we discover in Fig. 1 that AM (ctrl-AM) works significantly better than its DM counterpart (ctrl-DM).

**Further Connection to Target Matching**   Such contrast resembles the distinction between **target matching (TM)** and **denoising matching (DM)** in the literature of learning

*Table 2.* Learning to sample from lattice Ising models with $L = 24$. **Best** and second best results among all *uniform-based* discrete neural samplers are highlighted. ∗: Wall-clock time measured on 2 NVIDIA RTX A6000 GPUs. Full runtime comparison is in Tab. 5. †: For $\beta_{\text{critical}}$ and $\beta_{\text{low}}$, using warm-up strategy in PDNS (Guo et al., 2026). ‡: Failed to converge to meaningful distributions at $\beta_{\text{critical}}$ and $\beta_{\text{low}}$ even with warm-up. DASBS metrics are averaged over three random seeds.

| | Inv. Temp. | | | $\beta_{\text{high}} = 0.28$ | | | $\beta_{\text{critical}} = 0.4407$ | | | $\beta_{\text{low}} = 0.6$ | | |
|---|---|---|---|---|---|---|---|---|---|---|---|---|
| Type | Metrics ↓ | Steps (×1e3)* | Runtime (h)* | ΔMag. | ΔCorr. | EW$_2$ | ΔMag. | ΔCorr. | EW$_2$ | ΔMag. | ΔCorr. | EW$_2$ |
| Uniform | **DASBS** | **3.0** | **0.2** | $1.5e-2$ | $2.3e-3$ | 5.4 | $5.7e-1$ | **$1.2e-1$** | **53.7** | **$9.7e-3$** | **$1.9e-3$** | **2.5** |
| | LEAPS | 30 | 8.4 | **$1.8e-3$** | **$9.2e-4$** | **3.1** | **$5.9e-2$** | $2.8e-1$ | 96.5 | $3.0e-2$ | $5.5e-1$ | 176.6 |
| | UDNS‡ | 50 | 11.9 | $9.0e-3$ | $8.7e-3$ | 23.6 | – | – | – | – | – | – |
| | DFNS‡ | 50 | 2.1 | $9.3e-1$ | $8.0e-1$ | 661.6 | – | – | – | – | – | – |
| Masked | MDNS† | 50 | 16.8 | $3.9e-3$ | $7.4e-4$ | 0.1 | $1.1e-2$ | $5.6e-3$ | 5.1 | $9.0e-3$ | $4.7e-3$ | 5.3 |
| MCMC | MH | – | – | $8.9e-4$ | $2.9e-4$ | 1.2 | $2.5e-2$ | $3.7e-3$ | 293.3 | $4.0e-2$ | $6.6e-4$ | 109.9 |

scores of continuous probability distributions (De Bortoli et al., 2024; Kahouli et al., 2025): for two continuous random vectors $x, y \sim p(x, y)$, we can express the score as

$$\underbrace{\mathbb{E}_{p(x|y)} \nabla_x \log p(x)}_{\text{for additive } p(y|x)} = \nabla_y \log p(y) = \underbrace{\mathbb{E}_{p(x|y)} \nabla_y \log p(y|x)}_{\text{for general } p(y|x)},$$

where **additive** means $p(y|x) = q(y - x)$ for some distribution $q$. In other words, TM regresses onto the target score, whereas DM regresses onto the score of the transition kernel. TM leverages a more meaningful learning signal and avoids the numerical instability of DM when the noise level is very small (where $\nabla_y \log p(y|x)$ becomes singular), thereby providing faster convergence of training, a benefit we observe directly in our discrete experiments (Fig. 1).

**Unified View of Adjoint Matching** Adjoint matching (**AM**, Domingo-Enrich et al. (2025a)) is originally derived by analyzing the ODE of the *adjoint state* – the gradient of the cost-to-go from $X_t = x$ with respect to $x$. This formulation connects the adjoint state to the gradient of the objective with respect to control parameters, yielding a learning objective whose **unique fixed-point** corresponds to the optimal control. However, such gradient-based perspective is not directly generalizable to discrete domains. Recently, So et al. (2026) made a first attempt by expressing the optimal control as an expectation under the optimal path measure; however, their reliance on *masked* transition rates resulted in a complex training objective, due to the lack of the additive property. In contrast, our derivation identifies that the **additive reference noise** is the key structural requirement that enables a TM-like loss in discrete domains. We refer readers to the lower half of Tab. 1 for a side-by-side comparison of DM and AM formulations across continuous and discrete settings, and conclude with a unified characterization of the intrinsic nature of AM:

*Adjoint matching: a **fixed-point iteration** driven by a **target matching** objective converging to the **optimal** $p^{\star}$.*

**DASBS Unifies Existing Memoryless SOC Solvers** We further establish the connection between DASBS and the **weighted denoising cross-entropy (WDCE)** loss for solving *memoryless* SOC problems (Zhu et al., 2025a;b) (see Props. 6 and 7 for full statement and proof):

**Proposition 1.** *Under a* memoryless *reference path measure $p^r$ (encompassing both uniform (16) and masked (39) discrete diffusion), the denoising loss for the controller (ctrl-DM), with trajectory importance reweighting, generalized KL divergence, [2] and time weight $w_t \leftarrow \frac{\gamma_t}{N}$, is equivalent to the WDCE loss.*

**Convergence Analysis** Finally, following the continuous arguments (Liu et al., 2025a, Thm. 4), we establish the following convergence guarantee of DASBS.

**Theorem 3.** *(1) The path measure induced by the unique fixed-point of (ctrl-AM) ($\Phi^{(k)}$) solves a **forward half bridge problem** $\min_{p \text{ s.t. } p_0 = \mu} \text{KL}(p \| q^{\widehat{\varphi}_1^{(k-1)}})$ for some path measure $q^{\widehat{\varphi}_1^{(k-1)}}$ induced by $\widehat{\varphi}_1^{(k-1)}$ (see Def. 1).*

*(2) The unique fixed-point of (corr-DM) is the same as the unique fixed-point of (corr-AM). Denote it as $\widehat{\Phi}^{(k)}$, which induces a path measure $q^{\widehat{\varphi}_1^{(k)}}$ that solves a **backward half bridge problem** $\min_{q \text{ s.t. } q_1 = \nu} \text{KL}(p^{r\Phi^{(k)}} \| q)$.*

Consequently, the resulting update is alternatively projecting the current path measure onto the two constraint sets defined by the optimal controller and corrector, and the convergence is guaranteed by the theory of iterative proportional fitting (Rüschendorf, 1995; Chen et al., 2016a; De Bortoli et al., 2021). See Sec. C.11 for the proof.

# 6. Experiments

In this section, we evaluate the effectiveness and efficiency of the proposed DASBS algorithm. We demonstrate that it achieves competitive sample quality across standard benchmarks while offering significant advantages in training

---

[2] $f(t) = t \log t \implies D_f(a \| b) = a \log \frac{a}{b} - a + b$.

*Table 3.* Learning to sample from lattice Potts models with $L = 16$ and $N = 4$. **Best** results among all *uniform-based* discrete neural samplers are highlighted. †: For $\beta_{\text{critical}}$ and $\beta_{\text{low}}$, using warm-up strategy in PDNS (Guo et al., 2026). Full runtime comparison is in Tab. 6. DASBS numbers are averaged over three random seeds.

| | Inv. Temp. | $\beta_{\text{high}} = 0.9$ | | | $\beta_{\text{critical}} = 1.0986$ | | | $\beta_{\text{low}} = 1.3$ | | |
|---|---|---|---|---|---|---|---|---|---|---|
| Type | Metrics ↓ | $\Delta$Mag. | $\Delta$Corr. | $\text{EW}_2$ | $\Delta$Mag. | $\Delta$Corr. | $\text{EW}_2$ | $\Delta$Mag. | $\Delta$Corr. | $\text{EW}_2$ |
| Uniform | **DASBS** | $1.4e-2$ | $8.2e-3$ | **6.6** | **$4.2e-2$** | **$3.2e-2$** | **14.6** | **$1.1e-3$** | **$2.1e-3$** | **1.4** |
| | LEAPS | **$5.7e-3$** | **$5.0e-3$** | 8.2 | $3.2e-1$ | $2.6e-1$ | 79.9 | $3.6e-1$ | $3.5e-1$ | 90.5 |
| Mask | MDNS | $6.5e-3$ | $5.1e-3$ | 6.4 | $5.2e-3$ | $4.6e-3$ | 1.9 | $8.4e-4$ | $6.1e-4$ | 0.7 |
| MCMC | MH | $5.2e-2$ | $4.7e-2$ | 98.6 | $4.9e-1$ | $4.0e-1$ | 273.2 | $6.8e-1$ | $6.4e-1$ | 313.1 |

speed. We further investigate the benefits of AM and non-memoryless reference dynamics through ablation studies. Additional details and results are provided in Sec. D.

**Experimental Setup**   We test our method on two discrete distributions from statistical physics: the Ising and Potts models on square lattices with periodic boundary conditions. They are well known for exhibiting phase transitions as a function of the inverse temperature (Onsager, 1944; Beffara & Duminil-Copin, 2012) and serve as standard benchmarks for discrete sampling. For the Ising model, we use a lattice of size $L = 24$ (i.e., sequence length $D = L^2 = 576$) with states $\{\pm 1\}$, and consider inverse temperatures $\beta_{\text{high}} = 0.28$, $\beta_{\text{critical}} = \log(1 + \sqrt{2})/2 \approx 0.4407$, and $\beta_{\text{low}} = 0.6$. For the Potts model, we use $L = 16$ ($D = L^2 = 256$) with $N = 4$ states, and consider $\beta_{\text{high}} = 0.9$, $\beta_{\text{critical}} = \log(1 + \sqrt{q}) \approx 1.0986$, and $\beta_{\text{low}} = 1.3$. For both distributions, the first-order oracle is analytically tractable and computed in a highly vectorized manner as shown in (40). We benchmark DASBS against leading discrete neural samplers, including LEAPS (Holderrieth et al., 2025), DFNS (Ou et al., 2025b), UDNS (Zhu et al., 2025a, App. F), and MDNS (Zhu et al., 2025a), as well as the classical Metropolis-Hastings (MH) algorithm. Unless stated otherwise, we use $\mu = p_{\text{unif}}$ and (corr-AM) for $\beta_{\text{high}}$, $\mu$ being zero-temperature distribution and (ctrl-DM) for $\beta_{\text{critical}}, \beta_{\text{low}}$; see Sec. D.2 for full training details.

**Results and Discussion**   Quantitative results for the Ising and Potts models are summarized in Tabs. 2 and 3, respectively. We assess sample quality using three metrics: the deviation of the magnetization ($\Delta$Mag.), the deviation of the 2-point correlation ($\Delta$Corr.), and the energy Wasserstein-2 distance ($\text{EW}_2$), all relative to ground-truth samples from the Swendsen-Wang (SW, Swendsen & Wang (1986; 1987)) algorithm. For the Ising model with high temperature, we also report the number of training steps and runtime on two A6000 GPUs. Across all regimes, DASBS delivers **satisfactory sample fidelity** that is highly competitive with state-of-the-art uniform-based discrete neural samplers. Notably, DASBS confers a distinct advantage in **training efficiency**.

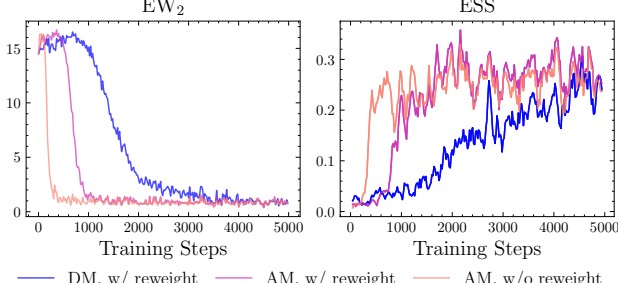

*Figure 1.* Ablation study of the adjoint matching (AM) and denoising matching (DM) training losses for the *memoryless* noise schedule $\gamma_t = \frac{1}{t}$ on Potts model with $L = 8$, $N = 3$, $\beta_{\text{high}} = 0.5$. Reweighting means using trajectory importance weight $p^\star/p^{\text{sg}(u)}$ in the training losses. DM with reweighting corresponds to UDNS (Zhu et al., 2025a, App. F). *Left*: Energy Wasserstein-2 distance to the ground-truth samples from SW algorithm. *Right*: Effective sample size computed from the trajectory importance weights.

We attribute this speed-up to three factors: (1) the use of the discrete score as a highly informative first-order oracle; (2) a memory-efficient design that avoids storing full roll-outs or computing loss over full trajectories; and (3) the simplicity of the matching loss objective.

**Ablation Study: AM v.s. DM in Learning Controller** In Fig. 2, we empirically validate the claim that AM is more efficient than DM. We focus on an $8 \times 8$ Potts model ($N = 3$, $\beta_{\text{high}} = 0.5$). We choose the standard *memoryless* noise schedule $\gamma_t = \frac{1}{t}$ (Ou et al., 2025a; Zhu et al., 2025a) to isolate the impact of the training objectives (ctrl-AM) v.s. (ctrl-DM). Fig. 1 shows that, while all three methods eventually converge, AM exhibits **significantly faster convergence** than DM when trajectory importance reweighting is enabled. Furthermore, we observe that the importance reweighting can inadvertently slow down the convergence due to small effective sample sizes (even after convergence), likely stemming from the variance of the estimated RND.

**Ablation Study: Memoryless v.s. Non-Memoryless**   We investigate the impact of the noise schedule $\gamma_t$ in Fig. 2, and

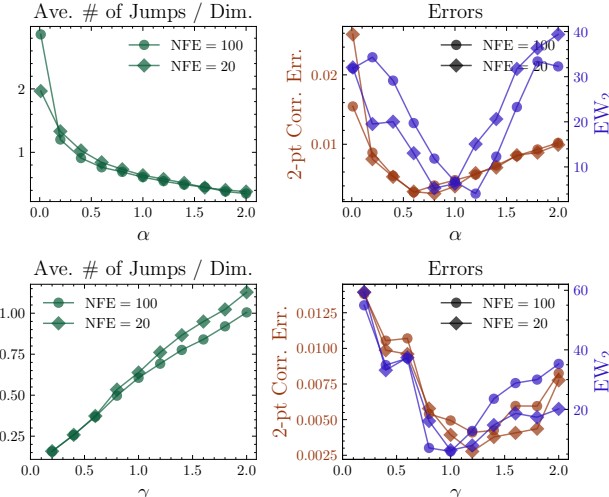

*Figure 2.* Ablation study of the hyperparameters $\alpha$ and $\gamma$ for the modified log-linear noise schedule $\gamma_t = \frac{\gamma}{t+\alpha}$ on Ising model with $L = 24$ and $\beta_{\text{high}} = 0.28$. The case of $\alpha = 0$ is *memoryless*. NFE is the number of function evaluations during generation for both training and inference. *Top*: fix $\gamma = 1$ and vary $\alpha$. *Bottom*: fix $\alpha = 1$ and vary $\gamma$. *Left*: average number of jumps for each dimension during generation. *Right*: 2-point correlation error and energy Wasserstein-2 distance to ground-truth samples drawn from SW algorithm.

in particular, we show that non-memoryless noise schedule performs better than the memoryless one. We adopt the *modified log-linear schedule* $\gamma_t = \frac{\gamma}{t+\alpha}$ on the $24 \times 24$ Ising model with $\beta_{\text{high}} = 0.28$, which is memoryless when $\alpha = 0$. Fig. 2 shows that when either hyperparameter is fixed at 1, the error metrics exhibit a distinct U-shaped curve as a function of the other parameter. Deviating from the optimal regime leads to degradation in sample quality: small $\gamma$ or large $\alpha$ reduces the transition rate magnitude and may hinder exploration, while large $\gamma$ or small $\alpha$ (particularly the memoryless case $\alpha = 0$) induces excessive jumps during generation, also degrading performance. Notably, these trends remain consistent across different computational budgets, demonstrating the robustness of the optimal configuration. A similar study for the constant noise schedule $\gamma_t \equiv \gamma$ is provided in Sec. D.3.

## 7. Conclusion and Future Work

In this work, we introduced a unified framework for discrete SB and SOC, proposing DASBS as an optimal-policy fixed-point iteration. By leveraging an additive noise scheme based on group structure, DASBS effectively extends AM to discrete domains. Several limitations remain: Like existing discrete neural samplers (Holderrieth et al., 2025; Zhu et al., 2025a), DASBS also exhibits potential training instability when the distribution has significant mode barriers (e.g., under critical and low temperatures). Our evaluation

is currently restricted to synthetic benchmarks, and performance on more complex distributions remains unknown. Additionally, the first-order nature of DASBS may be costly to implement if computing energy is expensive or not parallelizable. Future directions include extending the framework to discrete SOC with running costs (Domingo-Enrich et al., 2025a), exploring non-uniform reference dynamics like the Ehrenfest process (Winkler et al., 2024), and broadening our theoretical insights into AM to general state (e.g., Park et al. (2024); Woo et al. (2025); Park et al. (2025)).

## Acknowledgments

We thank the anonymous reviewer fjD9 for providing constructive comments regarding free energy estimation under the proposed framework. The authors are grateful for partial supports by NSF Grants ECCS-1942523, DMS-2206576, 2450378 (WG & YC), AFOSR Grant FA9550-25-1-0169 (WG & YC), Georgia Tech ARC-ACO Fellowship (WG), NSF Grant DMS-2513699 (YZ & MT), DOE Grants NA0004261 (MT), SC0026274 (YZ & MT), Richard Duke Fellowship (YZ & MT), Simons Institute for the Theory of Computing at UC Berkeley (MT), Amazon funding as part of the MIT Climate and Sustainability Consortium (MCSC) (XD), and Mathworks Fellowship (JN).

## Impact Statement

This paper presents work whose goal is to advance the field of Machine Learning. There are many potential societal consequences of our work, none of which we feel must be specifically highlighted here.

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

---

**Algorithm 1** Training of discrete adjoint Schrödinger bridge sampler (DASBS).

---

**Require:** Learnable controller $\Phi_t(x) \in \mathbb{R}_+^{D \times N}$ and corrector $\widehat{\Phi}(x) \in \mathbb{R}_+^{D \times N}$; reference CTMC with kernels $p_{t|0,1}^r$, $p_{1|t}^r$;
    initial distribution $\mu$; first-order oracle of $\nu$; Bregman divergence $D_f$; time weight $w_t$; inner step counts $S_{\text{ctrl}}, S_{\text{corr}}$;
    total epochs $S_{\text{epoch}}$.

1: Initialize $\widehat{\Phi}(x) \equiv 1$ (all-one corrector, Prop. 8 when applicable).
2: **for** epoch $k = 1, 2, \ldots, S_{\text{epoch}}$ **do**
3:     **[Controller training using (ctrl-AM) loss]** {freeze $\widehat{\Phi}$ via sg($\cdot$) when building targets}
4:     **for** controller step $s = 1$ **to** $S_{\text{ctrl}}$ **do**
5:         Sample $t \sim \text{Unif}(0,1)$, $(x_0, x_1) \sim p_{0,1}^{\text{sg}(r\Phi)}$, and $x \sim p_{t|0,1}^r(\cdot \mid x_0, x_1)$.
6:         For each $d \in [D]$ and $n \neq x^d$, set AM target $T_{d,n}^{\text{ctrl}} \leftarrow \varphi_1(x_1^{d \leftarrow x_1^d + n - x^d})/\varphi_1(x_1)$ using (21) with $\widehat{\Phi} \leftarrow \text{sg}(\widehat{\Phi})$ and
        $\varphi_1 = \nu/\widehat{\varphi}_1$.
7:         Update $\Phi$ by descending $w_t \sum_{d=1}^D \sum_{n \neq x^d} D_f\left(T_{d,n}^{\text{ctrl}} \middle\| \Phi_t(x)_{d,n}\right)$.
8:     **end for**
9:     **if** using corrector adjoint matching **then**
10:         **[Corrector training using (corr-AM) loss]** {freeze $\Phi$ via sg($\cdot$) when building targets}
11:         **for** corrector step $s = 1$ **to** $S_{\text{corr}}$ **do**
12:             Sample $(x_0, x_1) \sim p_{0,1}^{\text{sg}(r\Phi)}$.
13:             For each $d \in [D]$ and $n \neq x_1^d$, set target $T_{d,n}^{\text{corr}} \leftarrow \widehat{\varphi}_0(x_0^{d \leftarrow x_0^d + n - x_1^d})/\widehat{\varphi}_0(x_0)$ via (24) with $\Phi \leftarrow \text{sg}(\Phi)$.
14:             Update $\widehat{\Phi}$ by descending $\sum_{d=1}^D \sum_{n \neq x_1^d} D_f\left(T_{d,n}^{\text{corr}} \middle\| \widehat{\Phi}(x_1)_{d,n}\right)$.
15:         **end for**
16:     **else**
17:         **[Corrector training using (corr-DM) loss]** {no additivity of $p_{1|t}^r$ required; use when $\mu$ is not fully supported, *cf.*
        Sec. 6}
18:         **for** corrector step $s = 1$ **to** $S_{\text{corr}}$ **do**
19:             Sample $t \sim \text{Unif}(0,1)$, $(x_0, x_1) \sim p_{0,1}^{\text{sg}(r\Phi)}$, and $x \sim p_{t|0,1}^r(\cdot \mid x_0, x_1)$.
20:             Update $\widehat{\Phi}$ by descending $w_t \sum_{d=1}^D \sum_{n \neq x_1^d} D_f\left(\frac{p_{1|t}^r(x_1^{d \leftarrow n} \mid x)}{p_{1|t}^r(x_1 \mid x)} \middle\| \widehat{\Phi}(x_1)_{d,n}\right)$.
21:         **end for**
22:     **end if**
23: **end for**
**output** Trained controller $\Phi$ and corrector $\widehat{\Phi}$.

---

## A. Related Works

**1. Adjoint Matching** The idea of AM (Domingo-Enrich et al., 2025a) can be traced back to the adjoint method in optimal control theory (Pontryagin, 1987), and a few of its earlier applications in machine learning (Han & E, 2016; Chen et al., 2018; Li et al., 2020; Domingo-Enrich et al., 2024). The core idea is to define the adjoint state $a_t(X; u)$ associated with a control $u$ and trajectory $X = (X_t)_{t \in [0,1]}$ as $\nabla_{X_t} J_t(X)$, where $J_t(X)$ is the cost-to-go along the trajectory starting from $X_t$. Then, one can establish the dynamics of the adjoint state as an ODE backward in time starting from $a_1$ being the gradient of the terminal cost. Furthermore, the gradient of the full cost-to-go at the initial time $t = 0$ with respect to the control parameter can be written as an integral associated with the adjoint state, thus leading to a matching objective using the stop-gradient operator. It is shown in Domingo-Enrich et al. (2025a) that the optimal control is the unique fixed-point of the learning objective. AM-based methods have been applied to multiple tasks including continuous neural sampler training (Havens et al., 2025; Liu et al., 2025a; Choi et al., 2025; Blessing et al., 2025a), fine-tuning diffusion/flow-based models (Blessing et al., 2025a; Liu et al., 2025b; Domingo-Enrich et al., 2025b), and transition path sampling (Pidstrigach et al., 2025; Howard et al., 2025), showing competitive performance.

While AM is popularly applied in continuous state spaces, its generalization to discrete state spaces remains unexplored. Recently, So et al. (2026) made a first attempt by defining the discrete adjoint state as an estimator of the exponential of the value difference, and establishing its connection with the optimal transition rate, leading to a matching objective whose

unique fixed-point is the optimal transition rate. However, their training loss requires rollouts from intermediate states given the samples in the buffer, which is complicated. Concurrent to our work, Domingo-Enrich & Han (2026) provided an interpretation of adjoint matching through the stochastic maximum principle. Whether this perspective extends to general state spaces is an interesting future direction.

**2. Discrete Diffusion Neural Samplers** Earlier explorations (Wu et al., 2019; Nicoli et al., 2020) leverage autoregressive neural samplers to learn to sample from statistical mechanical distributions. Later, the study of continuous neural samplers has inspired the construction of similar algorithms for learning discrete distributions. For uniform-based discrete diffusion, Sanokowski et al. (2025a) introduced a framework for discrete diffusion samplers and proposed forward and reverse KL training objectives; Holderrieth et al. (2025) employed the discrete Jarzynski equality and learned the transport via locally equivariant neural networks, Ou et al. (2025b) followed and refined this line of study by proposing a different loss formulation, and Kholkin et al. (2025) proposed a neural sampler based on target concrete score identity. For mask-based discrete diffusion, Zhu et al. (2025a) formulated the sampling problem as an SOC problem similar to Zhang & Chen (2022), and proposed samplers based on masked and uniform discrete diffusion. Concurrent to our work, Du et al. (2026) proposed MetaDNS) that integrates tempered metadynamics into discrete diffusion samplers. Finally, discrete diffusion neural samplers are also used for solving combinatorial optimization problems via sampling from a low-temperature target distribution, e.g., Sanokowski et al. (2023; 2024; 2025a); Ou et al. (2025b); Guo et al. (2026).

# B. Cyclic Group Structure and General Discrete Spaces

In this section, we provide more details and discussions on the cyclic group structure and possible extensions to general discrete spaces.

**Shift Invariance and Algebraic Closure** Continuous adjoint matching builds on additive reference noise, i.e., kernels of the form $p_{1|t}^r(y|x) = q_t(y - x)$. To port the same mechanism to discrete spaces we require an analogous *shift invariance* of the reference transition kernel (17), which is what enables the AM identities such as (22) and the practical objectives (ctrl-AM) and (corr-AM). Viewing $[N]^D$ merely as a box inside $\mathbb{Z}^D$ is not algebraically convenient: unconstrained integer shifts do not stay in $[N]^D$. Identifying each coordinate with the cyclic group $\mathbb{Z}_N$ provides the closure needed to speak of additive noise on the entire state space without boundary artifacts.

**Decoupling from the Target and from Semantics** The cyclic identification is tied to the reference dynamics, *not* to the target distribution or to any intrinsic geometry of the labels. States may encode purely categorical quantities (e.g., node types or vocabulary indices) or ordinal ones (e.g., counts); neither interpretation forces a physical "wrap-around" on the data. All we need is a fixed bijection between each coordinate's label set and $\mathbb{Z}_N$ so that the group operations in our derivations are well-defined.

**Product Spaces and Structured Domains** The construction extends beyond the case of $[N]^D$: any finite product $\mathcal{X} = \prod_{d=1}^D X_d$ can be written, after choosing bijections $X_d \cong [N_d]$, as $\prod_{d=1}^D \mathbb{Z}_{N_d}$. The same reference-rate (16) and additive-noise viewpoint go through with straightforward bookkeeping when the alphabet sizes $N_d$ vary across dimensions. For instance, **molecular graphs** on up to $V$ vertices can be encoded as $\mathcal{A}^V \times \mathcal{B}^{V(V-1)/2}$ with atom set $\mathcal{A}$ and bond set $\mathcal{B}$, so the framework applies in principle at the level of the structured state space; integrating it with geometric architectures (e.g., equivariant GNNs) and hard constraints such as valency-aware moves remains a promising direction for future work.

# C. Theoretical Derivation

### C.1. Kolmogorov Equations for Continuous-time Markov Chains

**Proposition 2.** *Suppose $p_{t|s}(y|x)$, $0 \leq s < t \leq 1$ are the transition kernels of a CTMC with transition rate $r$. Then the following equations hold:*

- *Kolmogorov forward equation (KFE):*

$$\partial_t p_{t|s}(y|x) = \sum_z r_t(y, z) p_{t|s}(z|x), \tag{KFE}$$

- *Kolmogorov backward equation (KBE):*

$$\partial_s p_{t|s}(y|x) = -\sum_z r_s(z,x)p_{t|s}(y|z). \tag{KBE}$$

These are standard results in the theory of CTMCs and can be proved by leveraging the definition of the transition rate.

### C.2. Schrödinger Bridge: Proof of Thm. 1

We refer readers to Léonard (2014) for a comprehensive review of the SB theory. For completeness, we provide a self-contained proof here.

**Lemma 1.** *There exist some non-negative functions $\phi, \varphi_1$ such that the optimal path measure $p^\star$ satisfies*

$$p^\star_{(0,1)|0,1} = p^r_{(0,1)|0,1}, \tag{26}$$

$$p^\star_{0,1}(x,y) = p^r_{0,1}(x,y)\phi(x)\varphi_1(y), \quad s.t. \ p^\star_0 = \mu, \ p^\star_1 = \nu. \tag{27}$$

*In other words, $p^\star(X_{[0,1]}) = p^r(X_{[0,1]})\phi(X_0)\varphi_1(X_1)$.*

*Proof.* By the chain rule of KL divergence, we have

$$\mathrm{KL}(p^u\|p^r) = \mathrm{KL}(p^u_{0,1}\|p^r_{0,1}) + \mathbb{E}_{p^u_{0,1}(x_0,x_1)}\mathrm{KL}(p^u_{(0,1)|0,1}(\cdot|x_0,x_1)\|p^r_{(0,1)|0,1}(\cdot|x_0,x_1)).$$

Therefore, the optimal $p^u_{(0,1)|0,1}$ is $p^r_{(0,1)|0,1}$, and we only need to solve the following *static* SB problem:

$$\min_{p_{0,1}} \mathrm{KL}(p_{0,1}\|p^r_{0,1})$$

$$\text{s.t. } p_0 = \mu, \ p_1 = \nu \iff \sum_y p_{0,1}(x,y) = \mu(x), \ \sum_x p_{0,1}(x,y) = \nu(y).$$

Let the Lagrangian multiplier functions for the above constraints be $\alpha(x)$ and $\beta(y)$, respectively. The Lagrangian function is

$$L(p_{0,1}, \alpha, \beta) = \sum_{x,y} p_{0,1}(x,y)\log\frac{p_{0,1}(x,y)}{p^r_{0,1}(x,y)}$$

$$+ \sum_x \alpha(x)\left(\sum_y p_{0,1}(x,y) - \mu(x)\right) + \sum_y \beta(y)\left(\sum_x p_{0,1}(x,y) - \nu(y)\right).$$

Setting the partial derivative with respect to $p_{0,1}(x,y)$ to zero, we have

$$\log\frac{p_{0,1}(x,y)}{p^r_{0,1}(x,y)} + 1 + \alpha(x) + \beta(y) = 0 \implies p^\star_{0,1}(x,y) = p^r_{0,1}(x,y)\mathrm{e}^{-1-\alpha(x)-\beta(y)}.$$

Therefore, by defining $\phi(x) := \mathrm{e}^{-1-\alpha(x)}$ and $\varphi_1(y) := \mathrm{e}^{-\beta(y)}$, we complete the proof. Note that $\phi$ and $\varphi_1$ are defined up to a constant scaling factor. □

**Lemma 2.** *Define $\widehat{\varphi}_0 := \phi p^r_0$, and define the SB potentials $\varphi_t$ and $\widehat{\varphi}_t$ through the following relation:*

$$\varphi_t(x) = \sum_y p^r_{1|t}(y|x)\varphi_1(y), \quad \widehat{\varphi}_t(x) = \sum_y p^r_{t|0}(x|y)\widehat{\varphi}_0(y). \tag{28}$$

*Then,*

$$\partial_t \varphi_t(x) = -\sum_y \varphi_t(y)r_t(y,x), \tag{29}$$

$$\partial_t \widehat{\varphi}_t(x) = \sum_y \widehat{\varphi}_t(y)r_t(x,y), \tag{30}$$

*and furthermore, (11) and (12) hold.*

*Proof.* First, we prove (29) and (30):

$$\partial_t \varphi_t(x) = \sum_y \partial_t p^r_{1|t}(y|x)\varphi_1(y) \underset{\text{(KBE)}}{=} -\sum_y \sum_z r_t(z,x)p^r_{1|t}(y|z)\varphi_1(y) = -\sum_z r_t(z,x)\varphi_t(z),$$

$$\partial_t \widehat{\varphi}_t(x) = \sum_y \partial_t p^r_{t|0}(x|y)\widehat{\varphi}_0(y) \underset{\text{(KFE)}}{=} \sum_y \sum_z r_t(x,z)p^r_{t|0}(z|y)\widehat{\varphi}_0(y) = \sum_z r_t(x,z)\widehat{\varphi}_t(z).$$

Next, we prove (11) and (12) by verifying that the partial derivatives with respect to $t$ in (11) and with respect to $s$ in (12) are zero:

$$\sum_y \partial_t(p^r_{t|s}(y|x)\varphi_t(y)) = \sum_y \sum_z r_t(y,z)p^r_{t|s}(z|x)\varphi_t(y) - \sum_y \sum_z p^r_{t|s}(y|x)r_t(z,y)\varphi_t(z) = 0,$$

$$\sum_y \partial_s(p^r_{t|s}(x|y)\widehat{\varphi}_s(y)) = -\sum_y \sum_z r_s(z,y)p^r_{t|s}(x|z)\widehat{\varphi}_s(y) + \sum_y \sum_z p^r_{t|s}(x|y)r_s(y,z)\widehat{\varphi}_s(z) = 0.$$

$\square$

*Proof of* (13). We first study the marginal distribution at an arbitrary time point $t \in [0,1]$:

$$p^\star_t(x) = \sum_{x_0,x_1} p^\star_{0,1}(x_0,x_1)p^\star_{t|0,1}(x|x_0,x_1) \underset{\text{(26) and (27)}}{=} \sum_{x_0,x_1} p^r_{0,1}(x_0,x_1)\phi(x_0)\varphi_1(x_1)p^r_{t|0,1}(x|x_0,x_1)$$

$$= \sum_{x_0} p^r_0(x_0)p^r_{t|0}(x|x_0)\phi(x_0) \sum_{x_1} p^r_{1|t}(x_1|x)\varphi_1(x_1) \underset{\text{(28)}}{=} \widehat{\varphi}_t(x)\varphi_t(x).$$

Next, we study the joint distribution at two arbitrary time points $0 \le s < t \le 1$:

$$p^\star_{s,t}(x,y) = \sum_{x_0,x_1} p^\star_{0,1}(x_0,x_1)p^\star_{s,t|0,1}(x,y|x_0,x_1)$$

$$\underset{\text{(26) and (27)}}{=} \sum_{x_0,x_1} p^r_{0,1}(x_0,x_1)\phi(x_0)\varphi_1(x_1)p^r_{s,t|0,1}(x,y|x_0,x_1)$$

$$= \sum_{x_0} p^r_{0,s,t,1}(x_0,x,y,x_1)\phi(x_0)\varphi_1(x_1)$$

$$= \sum_{x_0} p^r_0(x_0)p^r_{s|0}(x|x_0)\phi(x_0) \sum_{x_1} p^r_{t|s}(y|x)p^r_{1|t}(x_1|y)\varphi_1(x_1)$$

$$\underset{\text{(28)}}{=} \widehat{\varphi}_s(x)p^r_{t|s}(y|x)\varphi_t(y).$$

Therefore, the second and third equalities in (13) follow immediately.

*Proof of* (14). This is obvious from Lem. 1 and (13):

$$p^\star(X_{[0,1]}) = p^r(X_{[0,1]})\phi(X_0)\varphi_1(X_1) = p^r(X_{[0,1]})\frac{\widehat{\varphi}_0(X_0)}{\mu(X_0)}\varphi_1(X_1) = p^r(X_{[0,1]})\frac{\widehat{\varphi}_0(X_0)}{\mu(X_0)}\frac{\nu(X_1)}{\widehat{\varphi}_1(X_1)}.$$

$\square$

*Proof of* (10). From (13), for $y \ne x$,

$$u^\star_t(y,x) = \lim_{h\to 0} \frac{p^\star_{t+h|t}(y|x)}{h} = \lim_{h\to 0} \frac{p^r_{t+h|t}(y|x)}{h}\frac{\varphi_{t+h}(y)}{\varphi_t(x)} = \frac{\varphi_t(y)}{\varphi_t(x)}\lim_{h\to 0}\frac{p^r_{t+h|t}(y|x)}{h} = \frac{\varphi_t(y)}{\varphi_t(x)}r_t(y,x).$$

$\square$

## C.3. Stochastic Optimal Control: Proof of (15)

*Proof.* Using the chain rule of KL divergence and noting that $p_0^u = p_0^r = \mu$, we have

$$\mathrm{KL}(p^u \| p^r) + \mathbb{E}_{X \sim p^u} g(X_1) = \mathbb{E}_{\mu(X_0)} \left[ \mathbb{E}_{p^u(X_{(0,1]}|X_0)} \log \frac{p^u(X_{(0,1]}|X_0)}{p^r(X_{(0,1]}|X_0) \mathrm{e}^{-g(X_1)}} \right].$$

Therefore, for any $X_0$, the optimal $p^u(X_{(0,1]}|X_0)$ is

$$p^\star(X_{(0,1]}|X_0) = \frac{1}{Z(X_0)} p^r(X_{(0,1]}|X_0) \mathrm{e}^{-g(X_1)},$$

where

$$Z(X_0) = \mathbb{E}_{p^r(X_{(0,1]}|X_0)\mathrm{e}^{-g(X_1)}} = \mathbb{E}_{p_{1|0}^r(y|X_0)} \mathrm{e}^{-g(y)}.$$

Finally, combining this with $p_0^\star = p_0^r = \mu$ completes the proof. $\qquad\square$

## C.4. Proof of Thm. 2

*Proof.* See Secs. C.2 and C.3 for the proofs of (14) and (15), respectively. To conclude the proof, it suffices to show the equivalence when $g \leftarrow \log \frac{\widehat{\varphi}_1}{\nu}$:

$$Z(x) \underset{(15)}{=} \sum_y \frac{p_{1|0}^r(y|x)}{\widehat{\varphi}_1(y)} \nu(y) \underset{(13)}{=} \sum_y \frac{p_{0|1}^\star(x|y)}{\widehat{\varphi}_0(x)} \nu(y) = \sum_y \frac{p_{0,1}^\star(x,y)}{\widehat{\varphi}_0(x)} = \frac{\mu(x)}{\widehat{\varphi}_0(x)}.$$

$\qquad\square$

**Corollary 1.** *The problem* (SOC) *with terminal cost g is equivalent to the problem* (SB) *with terminal distribution*

$$\nu(x) = \mathrm{e}^{-g(x)} \sum_y p_{1|0}^r(x|y) \frac{\mu(y)}{Z(y)}, \qquad \text{where } Z(y) = \mathbb{E}_{p_{1|0}^r(\cdot|y)} \mathrm{e}^{-g}.$$

*Proof.* It suffices to express $\nu$ in terms of $g$ by comparing (14) and (15). We have $\mu = \widehat{\varphi}_0 Z$ and $\nu = \widehat{\varphi}_1 \mathrm{e}^{-g}$. From (12), $\widehat{\varphi}_1(x) = \sum_y p_{1|0}^r(x|y)\widehat{\varphi}_0(y)$. Combining all these three equations completes the proof. $\qquad\square$

## C.5. Uniform Reference Dynamics

We write $r_t(y,x) = \gamma_t r(y,x)$ where

$$r(y,x) = \begin{cases} \frac{1}{N}, & \text{if } d_\mathrm{H}(y,x) = 1, \\ -D\left(1 - \frac{1}{N}\right), & \text{if } y = x, \\ 0, & \text{if otherwise.} \end{cases}$$

By verifying the detailed balance condition, it is easy to see that this transition rate keeps $p_\mathrm{unif} = \mathrm{Unif}(\mathcal{X})$ invariant:

$$p_\mathrm{unif}(x) r_t(y,x) = p_\mathrm{unif}(y) r_t(x,y) = \frac{\gamma_t}{N^{D+1}} 1_{d_\mathrm{H}(y,x)=1}, \ \forall x \neq y.$$

**Proposition 3.** *Define* $\overline{\gamma}_{s,t} := \int_s^t \gamma_u \mathrm{d}u$. *Then for* $x, y \in \mathcal{X}$ *and* $0 \leq s < t \leq 1$, *the reference transition probability is*

$$p_{t|s}^r(y|x) = A(s,t)^{d_\mathrm{H}(x,y)} B(s,t)^{D-d_\mathrm{H}(x,y)}, \ \text{where } A(s,t) := \frac{1 - \mathrm{e}^{-\overline{\gamma}_{s,t}}}{N}, \ B(s,t) := \frac{1 + (N-1)\mathrm{e}^{-\overline{\gamma}_{s,t}}}{N}. \tag{31}$$

**Remark 1.** *As a corollary,* $p_{1|t}^r(y|x) = q_t(y-x)$ *where* $q_t(\varepsilon) = A(t,1)^{d_\mathrm{H}(\varepsilon,0)} B(t,1)^{D-d_\mathrm{H}(\varepsilon,0)}$.

*Proof.* Note that each dimension evolves independently with transition rate matrix $\gamma_t \boldsymbol{R}$ where $\boldsymbol{R} \in \mathbb{R}^{N \times N}$ has off-diagonal entries $\frac{1}{N}$ and diagonal entries $\frac{1}{N} - 1$, i.e., $\boldsymbol{R} = \frac{1}{N}\mathbf{1}\mathbf{1}^{\mathrm{T}} - \boldsymbol{I}$. For $x \in [N]$, the vector $\boldsymbol{p}^r_{t|s}(\cdot|x) := (p^r_{t|s}(y|x))_{y \in [N]}$ satisfies the Kolmogorov forward equation $\partial_t \boldsymbol{p}^r_{t|s}(\cdot|x) = \gamma_t \boldsymbol{R} \boldsymbol{p}^r_{t|s}(\cdot|x)$ with initial condition $\boldsymbol{p}^r_{s|s}(\cdot|x) = \boldsymbol{e}^x$, i.e., the one-hot vector with 1 at the $x$-th entry. Therefore, we have $\boldsymbol{p}^r_{t|s}(\cdot|x) = \mathrm{e}^{\overline{\gamma}_{s,t}\boldsymbol{R}}\boldsymbol{e}^x$.

The matrix exponential can be computed as follows:

$$\mathrm{e}^{s\mathbf{1}\mathbf{1}^{\mathrm{T}}} = \sum_{k=0}^{\infty} \frac{s^k}{k!}(\mathbf{1}\mathbf{1}^{\mathrm{T}})^k = \boldsymbol{I} + \sum_{k=1}^{\infty} \frac{s^k}{k!}N^{k-1}\mathbf{1}\mathbf{1}^{\mathrm{T}} = \boldsymbol{I} + \frac{\mathrm{e}^{sN}-1}{N}\mathbf{1}\mathbf{1}^{\mathrm{T}},$$

$$\implies \mathrm{e}^{\lambda \boldsymbol{R}} = \mathrm{e}^{\frac{\lambda}{N}\mathbf{1}\mathbf{1}^{\mathrm{T}} - \lambda \boldsymbol{I}} = \mathrm{e}^{-\lambda}\mathrm{e}^{\frac{\lambda}{N}\mathbf{1}\mathbf{1}^{\mathrm{T}}} = \mathrm{e}^{-\lambda}\left(\boldsymbol{I} + \frac{\mathrm{e}^\lambda - 1}{N}\mathbf{1}\mathbf{1}^{\mathrm{T}}\right) = \mathrm{e}^{-\lambda}\boldsymbol{I} + \frac{1 - \mathrm{e}^{-\lambda}}{N}\mathbf{1}\mathbf{1}^{\mathrm{T}}.$$

Therefore, the per-dimension transition probability for $x, y \in [N]$ is

$$p^r_{t|s}(y|x) = \begin{cases} \frac{1 - \mathrm{e}^{-\overline{\gamma}_{s,t}}}{N} =: A(s,t) & \text{if } y \neq x, \\ \frac{1 + (N-1)\mathrm{e}^{-\overline{\gamma}_{s,t}}}{N} =: B(s,t) & \text{if } y = x, \end{cases} \tag{32}$$

which implies the full transition probability for $x, y \in \mathcal{X}$ is (31).

$\square$

**Remark 2.** *Note that when $\overline{\gamma}_{0,1} = \infty$, we have $p^r_{1|0}(y|x) = \frac{1}{N^D}$ for all $x, y \in \mathcal{X}$, i.e., the reference dynamics is memoryless. Under the assumption that $p^r_0 = p_{\mathrm{unif}}$, $p^r_t = p_{\mathrm{unif}}$ and thus $p^r_{t|1}(\cdot|y)$ can be implemented as follows: for each entry of $y$, independently, with probability $1 - \mathrm{e}^{-\overline{\gamma}_{t,1}}$, replace it with a random state from $\mathrm{Unif}[N]$.*

**Proposition 4.** *Under the reference dynamics $p^r$, each dimension of $p^r_{t|0,1}(x|x_0, x_1)$ for $x, x_0, x_1 \in \mathcal{X}$ is independent, and the per-dimensional distribution is given by*

$$p^r_{t|0,1}(x|x_0, x_1) = \begin{cases} \text{if } x_0 \neq x_1 : & \begin{cases} \frac{A(0,t)A(t,1)}{A(0,1)}, & \text{if } x \notin \{x_0, x_1\}, \\ \frac{B(0,t)A(t,1)}{A(0,1)}, & \text{if } x = x_0, \\ \frac{A(0,t)B(t,1)}{A(0,1)}, & \text{if } x = x_1, \end{cases} \\ \text{if } x_0 = x_1 : & \begin{cases} \frac{A(0,t)A(t,1)}{B(0,1)}, & \text{if } x \neq x_0 = x_1, \\ \frac{B(0,t)B(t,1)}{B(0,1)}, & \text{if } x = x_0 = x_1, \end{cases} \end{cases} \tag{33}$$

*for $x, x_0, x_1 \in [N]$.*

*Proof.* Due to the Markov property of $p^r$, we have

$$p^r_{t|0,1}(x|x_0, x_1) = \frac{p^r_{0,t,1}(x_0, x, x_1)}{p^r_{0,1}(x_0, x_1)} = \frac{p^r_{t|0}(x|x_0)p^r_{1|t}(x_1|x)}{p^r_{1|0}(x_1|x_0)}.$$

Note that again, each dimension is independent under $p^r$. Using the per-dimension transition probability (32), one can easily obtain the desired result. $\square$

**Choice of Noise Schedule** $\gamma_t$    We consider the following noise schedules:

- Constant schedule: $\gamma_t \equiv \gamma > 0$, $\overline{\gamma}_{s,t} = \gamma(t - s)$.

- Modified log-linear schedule: $\gamma_t = \frac{\gamma}{t+\alpha}$ for some $\gamma, \alpha > 0$, $\overline{\gamma}_{s,t} = \gamma \log \frac{t+\alpha}{s+\alpha}$. A larger $\alpha$ means a stronger memory effect, and $\alpha = 0$ recovers the memoryless case.

The ablation studies of the noise schedules can be found at Figs. 2 and 3.

## C.6. Inference via $\tau$-leaping

**Proposition 5.** *Using $\tau$-leaping to discretize the controlled CTMC, the transition probability for any dimension $d$ is approximated as*

$$\Pr(X_{t+h}^d = n | X_t = x) \approx \begin{cases} \frac{\overline{\gamma}_{t,t+h}}{N} \Phi_t(x)_{d,n}, & \text{if } n \neq x^d, \\ 1 - \frac{\overline{\gamma}_{t,t+h}}{N} \sum_{n' \neq x^d} \Phi_t(x)_{d,n'}, & \text{if } n = x^d. \end{cases}$$

*The approximate transition probability can be computed as*

$$
\begin{aligned}
p_{t+h|t}^u(y|x) &\approx \prod_{d=1}^D \Pr(X_{t+h}^d = y^d | X_t = x) \\
&= \prod_{d=1}^D \left( \frac{\overline{\gamma}_{t,t+h}}{N} \Phi_t(x)_{d,y^d} \right)^{1_{y^d \neq x^d}} \left( 1 - \frac{\overline{\gamma}_{t,t+h}}{N} \sum_{n' \neq x^d} \Phi_t(x)_{d,n'} \right)^{1_{y^d = x^d}}.
\end{aligned} \tag{34}
$$

We summarize the sampling procedure in Alg. 2. Note that for uniform discrete diffusion models, other samplers such as uniformization (Chen & Ying, 2025; Ren et al., 2025a) and higher-order $\tau$-leaping (Ren et al., 2025b) can also be applied, which we leave for future work.

---

**Algorithm 2** Sampling of discrete adjoint Schrödinger bridge sampler (DASBS) via $\tau$-leaping

---

**Require:** Model $\Phi$, initial distribution $\mu$, time discretization $0 = t_0 < t_1 < ... < t_M = 1$.
1: Sample initial state $x_0 \sim \mu$.
2: **for** $i = 0$ to $M - 1$ **do**
3:    Query $\Phi_{t_i}(x_i) \in \mathbb{R}_+^{D \times N}$ and compute $\overline{\gamma}_{t_i, t_{i+1}}$.
4:    For each $d \in [D]$ (in parallel), independently sample

$$\Pr(x_{i+1}^d = n) = \begin{cases} \frac{\overline{\gamma}_{t_i, t_{i+1}}}{N} \Phi_{t_i}(x_i)_{d,n}, & \text{if } n \neq x_i^d, \\ 1 - \frac{\overline{\gamma}_{t_i, t_{i+1}}}{N} \sum_{n' \neq x_i^d} \Phi_{t_i}(x_i)_{d,n'}, & \text{if } n = x_i^d. \end{cases}$$

5: **end for**
**output** $x_M$ as the generated sample.

---

## C.7. Target Matching Loss

*Proof of* (23).

$$
\begin{aligned}
\widehat{\varphi}_1(z) &\overset{(12)}{=} \sum_{x \in \mathbb{Z}_N^D} p_{1|t}^r(z|x) \widehat{\varphi}_t(x) \overset{(17)}{=} \sum_{x \in \mathbb{Z}_N^D} p_{1|t}^r(z + \Delta | x + \Delta) \widehat{\varphi}_t(x) \\
&\overset{=}{_{x' \leftarrow x + \Delta}} \sum_{x' \in \mathbb{Z}_N^D} p_{1|t}^r(z + \Delta | x') \widehat{\varphi}_t(x' - \Delta) \overset{=}{_{\Delta \leftarrow y - z}} \sum_{x' \in \mathbb{Z}_N^D} p_{1|t}^r(y | x') \widehat{\varphi}_t(x' - y + z), \\
\implies \frac{\widehat{\varphi}_1(z)}{\widehat{\varphi}_1(y)} &= \sum_x p_{1|t}^r(y|x) \frac{\widehat{\varphi}_t(x - y + z)}{\widehat{\varphi}_1(y)} \overset{(13)}{=} \sum_x p_{t|1}^\star(x|y) \frac{\widehat{\varphi}_t(x - y + z)}{\widehat{\varphi}_t(x)}.
\end{aligned}
$$

$\square$

## C.8. Denoising Matching Loss

We first prove the denoising matching characterization of the controller:

$$\frac{\varphi_t(y)}{\varphi_t(x)} \underset{(11)}{=} \sum_z p^r_{1|t}(z|y) \frac{\varphi_1(z)}{\varphi_t(x)} = \sum_z \frac{p^r_{1|t}(z|y)}{p^r_{1|t}(z|x)} \frac{p^r_{1|t}(z|x)\varphi_1(z)}{\varphi_t(x)} \underset{(13)}{=} \sum_z \frac{p^r_{1|t}(z|y)}{p^r_{1|t}(z|x)} p^\star_{1|t}(z|x), \tag{35}$$

$$\implies \Phi^\star = \underset{\Phi}{\arg\min}\, \mathbb{E}_t \mathbb{E}_{p^\star_{t,1}(x,x_1)} \sum_{d=1}^D \sum_{n \neq x^d} D_f\left( \frac{p^r_{1|t}(x_1|x^{d\leftarrow n})}{p^r_{1|t}(x_1|x)} \middle\| \Phi_t(x)_{d,n} \right). \tag{36}$$

Thus, the denoising matching loss for the controller reads

$$\Phi^{(k)} := \underset{\Phi}{\arg\min}\, \mathbb{E}_t w_t \mathbb{E}_{\substack{p^{sg(r\Phi)}_{0,1}(x_0,x_1) \\ p^r_{t|0,1}(x|x_0,x_1)}} \sum_{d=1}^D \sum_{n \neq x^d} D_f\left( \frac{p^r_{1|t}(x_1|x^{d\leftarrow n})}{p^r_{1|t}(x_1|x)} \middle\| \Phi_t(x)_{d,n} \right). \tag{ctrl-DM}$$

Moreover, the variational characterization of $\widehat{\Phi}^\star$ in the style of denoising matching is as follows:

$$\widehat{\Phi}^\star = \underset{\widehat{\Phi}}{\arg\min}\, \mathbb{E}_t \mathbb{E}_{p^\star_{t,1}(x,x_1)} \sum_{d=1}^D \sum_{n \neq x^d} D_f\left( \frac{p^r_{1|t}(x^{d\leftarrow n}_1|x)}{p^r_{1|t}(x_1|x)} \middle\| \widehat{\Phi}(x_1)_{d,n} \right).$$

## C.9. Trajectory Importance Reweighting

By **trajectory importance reweighting**, we refer to the practice when we use the RND $\frac{p^\star(x_{[0,1]})}{p^u(x_{[0,1]})}$ in computing the losses (ctrl-AM) to (corr-DM). For instance, (ctrl-AM) becomes the following form:

$$\Phi^{(k)} := \underset{\Phi}{\arg\min}\, \mathbb{E}_t w_t \boxed{\mathbb{E}_{p^{sg(r\Phi)}(x_{[0,1]})} \frac{p^\star(x_{[0,1]})}{p^{sg(r\Phi)}(x_{[0,1]})}} \mathbb{E}_{p^r_{t|0,1}(x|x_0,x_1)} \sum_{d=1}^D \sum_{n \neq x^d} D_f\left( \frac{\varphi_1(x^{d\leftarrow x^d_1+n-x^d}_1)}{\varphi_1(x_1)} \middle\| \Phi_t(x)_{d,n} \right).$$

In practice, for stability, after obtaining the log RND $\log \frac{p^\star(x_{[0,1]})}{p^{sg(r\Phi)}(x_{[0,1]})}$ up to an additive constant over a batch of trajectories, one typically applies *softmax* to normalize the sum of weights to one.

Leveraging Props. 3 and 5, we can approximately compute $\log \frac{p^u(x_{[0,1]})}{p^r(x_{[0,1]})}$. With the approximation in the following Lem. 3, we can approximate the log weights for importance sampling, $\log \frac{p^\star(x_{[0,1]})}{p^u(x_{[0,1]})}$.

**Lemma 3.** *When the reference dynamics is memoryless,* $\log \frac{p^\star(x_{[0,1]})}{p^r(x_{[0,1]})} = \log \frac{\nu(x_1)}{p^r_1(x_1)} + \text{const}$; *otherwise, the log RND can be computed and approximated as follows:*

$$\log \frac{p^\star(x_{[0,1]})}{p^r(x_{[0,1]})} = \int_0^1 \sum_{y \neq x_t} \left(1 - \frac{\varphi_t(y)}{\varphi_t(x_t)}\right) r_t(y,x_t)\mathrm{d}t + \sum_{t:x_{t_-} \neq x_t} \log \frac{\varphi_t(x_t)}{\varphi_t(x_{t_-})} \tag{37}$$

$$\approx \sum_{i=0}^{M-1} \left[ \sum_{d=1}^D \sum_{n \neq x^d_{t_i}} \frac{\overline{\gamma}_{t_i,t_{i+1}}}{N} \left(1 - \Phi_{t_i}(x_{t_i})_{d,n}\right) + \sum_{d:x^d_{t_i} \neq x^d_{t_{i+1}}} \log \Phi_{t_i}(x_{t_i})_{d,x^d_{t_{i+1}}} \right], \tag{38}$$

*where* $0 = t_0 < ... < t_M = 1$ *are the discretized time points.*

*Proof.* From (14), one can find $\log \frac{p^\star(x_{[0,1]})}{p^r(x_{[0,1]})} = \log \frac{\varphi_1(x_1)}{\varphi_0(x_0)}$. When memoryless, following the argument in Sec. 3, $\varphi_0 = \text{const}$, $\log \varphi_1 = \log \frac{\nu}{\widetilde{\varphi}_1} = \log \frac{\nu}{p^r_1} + \text{const}$, which yields the desired result.

For general cases, inspired by the discussion in Liu et al. (2025a, App. D.4): (29) implies

$$\partial_t \varphi_t(x) = \sum_{y \neq x} (\varphi_t(x) - \varphi_t(y)) r_t(y,x) \implies \partial_t \log \varphi_t(x) = \sum_{y \neq x} \left(1 - \frac{\varphi_t(y)}{\varphi_t(x)}\right) r_t(y,x).$$

Hence, we have

$$\log \frac{\varphi_1(x_1)}{\varphi_0(x_0)} = \int_0^1 \partial_t \log \varphi_t(x_t)\mathrm{d}t + \sum_{t:x_{t_-} \neq x_t} \log \frac{\varphi_t(x_t)}{\varphi_t(x_{t_-})}$$

$$= \int_0^1 \sum_{y \neq x_t} \left(1 - \frac{\varphi_t(y)}{\varphi_t(x_t)}\right) r_t(y, x_t)\mathrm{d}t + \sum_{t:x_{t_-} \neq x_t} \log \frac{\varphi_t(x_t)}{\varphi_t(x_{t_-})}.$$

Under our setting, the summation in the first term can be easily calculated:

$$\sum_{y \neq x_t} \left(1 - \frac{\varphi_t(y)}{\varphi_t(x_t)}\right) r_t(y, x_t) = \sum_{d=1}^D \sum_{n \neq x_t^d} \left(1 - \frac{\varphi_t(x_t^{d \leftarrow n})}{\varphi_t(x_t)}\right) r_t(x_t^{d \leftarrow n}, x_t) = \sum_{d=1}^D \sum_{n \neq x_t^d} (1 - \Phi_t^\star(x_t)_{d,n}) \frac{\gamma_t}{N}.$$

For the second term, in theory, $x_{t_-}$ and $x_t$ differ by one dimension only, but during discretization, multiple dimensions may change simultaneously. A heuristic approximation is to decompose it into multiple single-dimension changes:

$$\log \frac{\varphi_t(x_t)}{\varphi_t(x_{t_-})} \approx \sum_{d:x_{t_-}^d \neq x_t^d} \log \frac{\varphi_t(x_{t_-}^{d \leftarrow x_t^d})}{\varphi_t(x_{t_-})} = \sum_{d:x_{t_-}^d \neq x_t^d} \log \Phi_t^\star(x_{t_-})_{d,x_t^d}.$$

We can thus summarize the approximate calculation of $\frac{p^\star(x_{[0,1]})}{p^r(x_{[0,1]})} = \frac{\varphi_1(x_1)}{\varphi_0(x_0)}$ on the time-discretized trajectory $(x_{t_0}, x_{t_1}, ..., x_{t_M})$ as follows:

$$\log \frac{\varphi_1(x_1)}{\varphi_0(x_0)} \approx \sum_{i=0}^{M-1} \left[\sum_{d=1}^D \sum_{n \neq x_{t_i}^d} \frac{\overline{\gamma}_{t_i, t_{i+1}}}{N} \left(1 - \Phi_{t_i}^\star(x_{t_i})_{d,n}\right) + \sum_{d:x_{t_i}^d \neq x_{t_{i+1}}^d} \log \Phi_{t_i}^\star(x_{t_i})_{d,x_{t_{i+1}}^d}\right].$$

Finally, as the ground-truth $\Phi^\star$ is unavailable, we use $\Phi$ to approximate its value. This concludes the proof.

$\square$

**Remark 3.** *Unlike the method proposed in* Liu et al. *(2025a, App. D.4), here we don't need to train $\frac{\widehat{\varphi}_t(y)}{\widehat{\varphi}_t(x)}$ along the trajectory. This is because here we derive through $\frac{\varphi_1(x_1)}{\varphi_0(x_0)}$ instead of $\frac{\widehat{\varphi}_1(x_1)}{\widehat{\varphi}_0(x_0)}$.*

**Remark 4.** *In the case of masked diffusion, the path measure $p^u$ can be exactly sampled and the log RND on a give trajectory can be computed precisely; however, here, the estimated $\log \frac{p^\star(x_{[0,1]})}{p^u(x_{[0,1]})}$ involves time discretization error, and furthermore, learning error if we use* (38) *under non-memoryless cases. This possibly explains the low ESS observed in* Fig. 1 *even after convergence.*

## C.10. Connection between DASBS and WDCE

**Proposition 6.** *Assume $\overline{\gamma}_{0,1} = \infty$, i.e., the reference path measure $p^r$ is memoryless. Then the denoising loss for training the controller* (ctrl-DM) *with trajectory importance reweighting, $D_f$ being the generalized KL divergence,[2] and time-weight $w_t \leftarrow \frac{\gamma_t}{N}$ reduces the denoising cross-entropy (WDCE) loss in UDNS (Zhu et al., 2025a, App. F), which is equal to $\mathrm{KL}(p^\star \| p^u) + \mathrm{const}$.*

*Proof.* The first equation on Zhu et al. (2025a, Page 39) reads (using the notation in this paper)

$$\mathrm{KL}(p^\star \| p^u) = \mathbb{E}_{p^{\mathrm{sg}(u)}(\overline{x}_{[0,1]})} \frac{p^\star(\overline{x}_{[0,1]})}{p^{\mathrm{sg}(u)}(\overline{x}_{[0,1]})} \mathbb{E}_t \frac{\gamma_t}{N} \mathbb{E}_{p_{t|1}^\star(x|\overline{x}_1)} \sum_{d=1}^D \sum_{n \neq x^d} D_f \left(\frac{p_{t|1}^\star(x^{d \leftarrow n}|\overline{x}_1)}{p_{t|1}^\star(x|\overline{x}_1)} \middle\| \Phi_t(x)_{d,n}\right) + \mathrm{const},$$

where $D_f$ is the generalized KL divergence. Under the memoryless assumption, we have $p_{t|1}^\star(x|x_1) = p_{t|1}^r(x|x_1)$. Due to the symmetry $p_{t|1}^r(x|x_1) = p_{t|1}^r(x_1|x)$, the equivalence to (ctrl-DM) is obvious.

$\square$

**Proposition 7.** *Consider the mask-augmented state space $\mathcal{X} = \{1, ..., N, \mathbf{M}\}^D$. Let the reference transition rate be*

$$r_t(y, x) = \begin{cases} \frac{\gamma_t}{N}, & \text{if } y = x^{d\leftarrow n}, \ x^d = \mathbf{M}, \ n \in [N], \\ -\gamma_t|\{d : x^d = \mathbf{M}\}|, & \text{if } y = x, \\ 0, & \text{otherwise,} \end{cases} \tag{39}$$

*for some noise schedule $\gamma_\cdot : [0, 1] \to \mathbb{R}_+$. Let $\overline{\gamma}_{s,t} := \int_s^t \gamma_u \mathrm{d}u$ for $0 \le s < t \le 1$, and assume $\overline{\gamma}_{t,1} = \infty$ for any $0 \le t < 1$. Then, the denoising loss for training the controller (ctrl-DM) with trajectory importance weighting, $D_f$ being the generalized KL divergence,[2] and time-weight $\frac{\gamma_t}{N}$ reduces to the WDCE loss in MDNS (Zhu et al., 2025a), which is also equal to $\mathrm{KL}(p^\star \| p^u) + \mathrm{const}$.*

*Proof.* Throughout the proof, we always assume $n \in [N]$ is a non-mask state. From Zhu et al. (2025a), we have the following results:

$$u_t^\star(x^{d\leftarrow n}, x) = \gamma_t \Pr_{X \sim \nu}(X^d = n | X^{\mathrm{UM}} = x^{\mathrm{UM}}) 1_{x^d = \mathbf{M}}$$

$$\Phi_t^\star(x)_{d,n} = \frac{\varphi_t(x^{d\leftarrow n})}{\varphi_t(x)} = N \Pr_{X \sim \nu}(X^d = n | X^{\mathrm{UM}} = x^{\mathrm{UM}}) 1_{x^d = \mathbf{M}}$$

$$p_{t|s}^r(y|x) = \prod_{d : x^d = \mathbf{M}} \left(\frac{1 - e^{-\overline{\gamma}_{s,t}}}{N}\right)^{1_{y^d \neq \mathbf{M}}} \left(e^{-\overline{\gamma}_{s,t}}\right)^{1_{y^d = \mathbf{M}}} \cdot \prod_{d : x^d \neq \mathbf{M}} 1_{x^d = y^d}, \ 0 \le s < t \le 1$$

$$\implies \frac{p_{1|t}^r(x_1|x^{d\leftarrow n})}{p_{1|t}^r(x_1|x)} = N 1_{x_1^d = n} 1_{x^d = \mathbf{M}} \ (\text{suppose } \forall d \text{ s.t. } x^d \neq \mathbf{M}, \ x_1^d = x^d)$$

Let $s_\theta : \mathcal{X} \to \mathbb{R}_{\ge 0}^{D \times N}$ be the neural network to learn the conditional distribution in $\nu$, i.e., $s_\theta(x)_{d,n} \approx \Pr_{X \sim \nu}(X^d = n | X^{\mathrm{UM}} = x^{\mathrm{UM}})$ for $x^d = \mathbf{M}$, and $s_\theta(x)_{d,n} = 1_{x^d = n}$ for $x^d \neq \mathbf{M}$. We assume $\sum_{n=1}^N s_\theta(x)_{d,n} = 1$ for all $d$.

Therefore, with generalized KL divergence, the loss (ctrl-DM) with trajectory importance reweighting becomes

$$\mathbb{E}_t w_t \mathbb{E}_{p^{\mathrm{sg}(\theta)}(x_{[0,1]})} \frac{\mathrm{d}p^\star}{\mathrm{d}p^{\mathrm{sg}(\theta)}}(x_{[0,1]}) \mathbb{E}_{p_{t|0,1}^r(x|x_0, x_1)} \sum_{d=1}^D \sum_{n \neq x^d} D_f(N 1_{x_1^d = n} 1_{x^d = \mathbf{M}} \| N s_\theta(x)_{d,n} 1_{x^d = \mathbf{M}} + 1_{x^d = n})$$

$$= \mathbb{E}_t w_t \mathbb{E}_{p^{\mathrm{sg}(\theta)}(x_{[0,1]})} \frac{\mathrm{d}p^\star}{\mathrm{d}p^{\mathrm{sg}(\theta)}}(x_{[0,1]}) \mathbb{E}_{p_{t|1}^r(x|x_1)} \sum_{d : x^d = \mathbf{M}} \sum_{n=1}^N D_f(N 1_{x_1^d = n} \| N s_\theta(x)_{d,n})$$

$$= \mathbb{E}_t w_t \mathbb{E}_{p^{\mathrm{sg}(\theta)}(x_{[0,1]})} \frac{\mathrm{d}p^\star}{\mathrm{d}p^{\mathrm{sg}(\theta)}}(x_{[0,1]}) \mathbb{E}_{p_{t|1}^r(x|x_1)} \sum_{d : x^d = \mathbf{M}} \sum_{n=1}^N \left(N 1_{x_1^d = n} \log \frac{1_{x_1^d = n}}{s_\theta(x)_{d,n}} - N 1_{x_1^d = n} + N s_\theta(x)_{d,n}\right)$$

$$= \mathbb{E}_t w_t \mathbb{E}_{p^{\mathrm{sg}(\theta)}(x_{[0,1]})} \frac{\mathrm{d}p^\star}{\mathrm{d}p^{\mathrm{sg}(\theta)}}(x_{[0,1]}) \mathbb{E}_{p_{t|1}^r(x|x_1)} \sum_{d : x^d = \mathbf{M}} -N \log s_\theta(x)_{d,x_1^d} + \mathrm{const}.$$

Thus, with $t \sim \mathrm{Unif}(0, 1)$ and time weights $\frac{\gamma_t}{N}$, the loss

$$\min_\theta \mathbb{E}_{p^{\mathrm{sg}(\theta)}(x_{[0,1]})} \frac{\mathrm{d}p^\star}{\mathrm{d}p^{\mathrm{sg}(\theta)}}(x_{[0,1]}) \mathbb{E}_{t \sim \mathrm{Unif}(0,1)} \gamma_t \mathbb{E}_{p_{t|1}^r(x|x_1)} \sum_{d : x^d = \mathbf{M}} -\log s_\theta(x)_{d,x_1^d},$$

which is exactly the same as the WDCE loss in Zhu et al. (2025a) if we choose the canonical noise schedule in masked diffusion model: $\gamma_t = \frac{1}{t}$. $\square$

## C.11. Convergence of Alternating Update: Proof of Thm. 3

**Definition 1.** *For a function $\psi : \mathcal{X} \to \mathbb{R}_+$, we use $q^\psi$ to denote the path measure of a CTMC $(Y_t)_{t\in[0,1]}$ induced by the* backward *transition rate $u_t^\leftarrow(y,x) = \frac{\psi_t(y)}{\psi_t(x)}r_t(x,y)$, $y \neq x$ and initialized at $Y_1 \sim \nu$, where $\psi_t$ satisfies $\psi_t(x) = \sum_y p_{t|0}^r(x|y)\psi_0(y)$, $\psi_1 = \psi$. In other words,*

$$u_t^\leftarrow(y,x) = \lim_{h\to 0} \frac{\Pr(Y_{t-h} = y|Y_t = x) - 1_{x=y}}{h}.$$

**Remark 5.** *By Bayes' rule, one can obtain its equivalent forward transition rate (Kelly, 2011): for $y \neq x$,*

$$
u_t^\rightarrow(y,x) = \lim_{h\to 0} \frac{1}{h}\Pr(Y_{t+h} = y|Y_t = x) = \lim_{h\to 0}\frac{1}{h}\frac{\Pr(Y_t = x|Y_{t+h} = y)\Pr(Y_{t+h} = y)}{\Pr(Y_t = x)}
$$

$$
= \lim_{h\to 0}\frac{1}{h}\frac{q_{t+h}^\psi(y)}{q_t^\psi(x)}(u_{t+h}^\leftarrow(x,y)h + o(h)) = \frac{q_t^\psi(y)}{q_t^\psi(x)}u_t^\leftarrow(x,y) = \frac{(q_t^\psi/\psi_t)(y)}{(q_t^\psi/\psi_t)(x)}r_t(y,x).
$$

### C.11.1. PROOF OF PART (1) OF THM. 3

*Proof.* Let $(Y_t)_{t\in[0,1]} \sim q^{\widehat{\varphi}_1^{(k-1)}} =: q$. By (KFE),

$$
\partial_t q_t(x) = \sum_y q_t(y)u_t^\rightarrow(x,y) = \sum_{y\neq x}(q_t(y)u_t^\rightarrow(x,y) - q_t(x)u_t^\rightarrow(y,x))
$$

$$
= \sum_{y\neq x}\left(\frac{(q_t/\psi_t)(x)}{(q_t/\psi_t)(y)}r_t(x,y)q_t(y) - \frac{(q_t/\psi_t)(y)}{(q_t/\psi_t)(x)}r_t(y,x)q_t(x)\right),
$$

$$
\implies \partial_t \log q_t(x) = \sum_{y\neq x}\left(\frac{\psi_t(y)}{\psi_t(x)}r_t(x,y) - \frac{(q_t/\psi_t)(y)}{(q_t/\psi_t)(x)}r_t(y,x)\right).
$$

On the other hand, using (KFE) again,

$$
\partial_t \psi_t(x) = \sum_y \partial_t p_{t|0}^r(x|y)\psi_0(y) = \sum_y \sum_z r_t(x,z)p_{t|0}^r(z|y)\psi_0(y) = \sum_z r_t(x,z)\psi_t(z)
$$

$$
= \sum_{y\neq x}(r_t(x,y)\psi_t(y) - r_t(y,x)\psi_t(x)),
$$

$$
\implies \partial_t \log \psi_t(x) = \sum_{y\neq x}\left(r_t(x,y)\frac{\psi_t(y)}{\psi_t(x)} - r_t(y,x)\right).
$$

Now we compute $\mathrm{KL}(p^u\|q)$ where $p^u$ is the path measure of a CTMC $(X_t)_{t\in[0,1]}$ induced by transition rate $u_t$ and initial distribution $\mu$:

$$
\mathrm{KL}(p^u\|q) = \mathrm{KL}(\mu\|q_0) + \mathbb{E}_{p^u(X)}\int_0^1 \sum_{y\neq X_t}\left(u_t \log\frac{u_t}{u_t^\rightarrow} + u_t^\rightarrow - u_t\right)(y,X_t)\mathrm{d}t
$$

$$
= \mathrm{KL}(\mu\|q_0) + \mathbb{E}_{p^u(X)}\int_0^1 \sum_{y\neq X_t}\left[\left(u_t \log\frac{u_t}{r_t} + r_t - u_t\right)(y,X_t)\right.
$$

$$
\left. + \left(-u_t(y,X_t)\log\frac{(q_t/\psi_t)(y)}{(q_t/\psi_t)(X_t)} + \frac{(q_t/\psi_t)(y)}{(q_t/\psi_t)(X_t)}r_t(y,X_t) - r_t(y,X_t)\right)\right]\mathrm{d}t.
$$

The second term is $\mathrm{KL}(p^u \| p^r)$. To deal with the third term, we leverage Lem. 4:

$$\mathbb{E}_{p^u(X)} \log \frac{(\psi_1/q_1)(X_1)}{(\psi_0/q_0)(X_0)} = \mathbb{E}_{p^u(X)} \Big[ \int_0^1 (\partial_t \log \psi_t(X_t) - \partial_t \log q_t(X_t)) \mathrm{d}t + \sum_{t: X_{t_-} \neq X_t} \log \frac{(\psi_t/q_t)(X_t)}{(\psi_t/q_t)(X_{t_-})} \Big]$$

$$= \mathbb{E}_{p^u(X)} \int_0^1 \sum_{y \neq X_t} \Big( \frac{(q_t/\psi_t)(y)}{(q_t/\psi_t)(X_t)} r_t(y, X_t) - r_t(y, X_t) \Big) \mathrm{d}t$$

$$+ \mathbb{E}_{p^u(X)} \int_0^1 \sum_{y \neq X_t} u_t(y, X_t) \log \frac{(\psi_t/q_t)(y)}{(\psi_t/q_t)(X_t)} \mathrm{d}t.$$

Therefore, we conclude that

$$\mathrm{KL}(p^u \| q) = \mathrm{KL}(p^u \| p^r) + \mathbb{E}_{p^u(X)} \log \frac{(\psi_1/q_1)(X_1)}{(\psi_0/q_0)(X_0)} + \mathrm{const}$$

$$= \mathrm{KL}(p^u \| p^r) + \mathbb{E}_{p^u(X)} \log \frac{\psi_1}{q_1}(X_1) + \mathrm{const}$$

$$= \mathrm{KL}(p^u \| p^r) + \mathbb{E}_{p^u(X)} \log \frac{\widehat{\varphi}_1^{(k-1)}}{\nu}(X_1) + \mathrm{const},$$

where $\mathrm{const}$ does not depend on $u$. Therefore, this is an SOC problem with terminal cost $g \leftarrow \log \frac{\widehat{\varphi}_1^{(k-1)}}{\nu}$. Let the optimal path measure to this SOC problem be $p^{(k)}$ and, by relating this SOC problem with an equivalent SB problem through Cor. 1, let the SB potentials to this problem be $(\varphi_t^{(k)}, \widehat{\varphi}_t^{(k)})$. Then, $\varphi_1^{(k)} = \mathrm{e}^{-g} = \frac{\nu}{\widehat{\varphi}_1^{(k-1)}}$ by Cor. 1. From Sec. C.2 and similar argument as (19), we can leverage the additive noise and obtain

$$\frac{\varphi_t^{(k)}(y)}{\varphi_t^{(k)}(x)} = \mathbb{E}_{p_{1|t}^{(k)}(x_1|x)} \frac{\varphi_1^{(k)}(x_1 + y - x)}{\varphi_1^{(k)}(x_1)} = \mathbb{E}_{p_{1|t}^{(k)}(x_1|x)} \frac{(\nu/\widehat{\varphi}_1^{(k-1)})(x_1 + y - x)}{(\nu/\widehat{\varphi}_1^{(k-1)})(x_1)}.$$

On the other hand, from the property of Bregman divergence, the unique fixed-point of (ctrl-AM), $\Phi^{(k)}$, satisfies

$$\Phi_t^{(k)}(x)_{d,n} = \mathbb{E}_{p_{1|t}^{r\Phi^{(k)}}(x_1|x)} \frac{(\nu/\widehat{\varphi}_1^{(k-1)})(x^{d \leftarrow x_1^d + n - x^d})}{(\nu/\widehat{\varphi}_1^{(k-1)})(x)}.$$

Thus, we conclude that $\Phi_t^{(k)}(x)_{d,n} = \frac{\varphi_t^{(k)}(x^{d \leftarrow n})}{\varphi_t^{(k)}(x)}$. $\qquad \square$

### C.11.2. PROOF OF PART (2) OF THM. 3

*Proof.* By similar arguments using Bayes formula, we can write $p^{r\Phi^{(k)}} =: p$ as a backward CTMC initialized at $p_1$ with backward transition rate

$$u_t^{\leftarrow(k)}(y, x) = \frac{(p_t^{(k)}/\varphi_t^{(k)})(y)}{(p_t^{(k)}/\varphi_t^{(k)})(x)} r_t(x, y) = \frac{\widehat{\varphi}_t^{(k)}(y)}{\widehat{\varphi}_t^{(k)}(x)} r_t(x, y),$$

where $\varphi_t^{(k)}$ and $\widehat{\varphi}_t^{(k)}$ are the SB potentials to the SOC problem in the proof of the first part. Then, for any backward CTMC $(Y_t)_{t \in [0,1]} \sim q$ initialized at $Y_1 \sim \nu$ with backward transition rate $u_t^{\leftarrow}$, we have

$$\mathrm{KL}(p \| q) = \mathrm{KL}(p_1 \| \nu) + \mathbb{E}_{X \sim p} \int_0^1 \sum_{y \neq X_t} \Big( u_t^{\leftarrow(k)} \log \frac{u_t^{\leftarrow(k)}}{u_t^{\leftarrow}} + u_t^{\leftarrow} - u_t^{\leftarrow(k)} \Big)(y, X_t) \mathrm{d}t.$$

Therefore, it is obvious that the optimal $u_t^{\leftarrow}(y, x)$ is equal to $u_t^{\leftarrow(k)}(y, x) = \frac{\widehat{\varphi}_t^{(k)}(y)}{\widehat{\varphi}_t^{(k)}(x)} r_t(x, y)$, i.e., the optimal $q$ is $q^{\widehat{\varphi}_1^{(k)}}$ (Def. 1). By similar arguments as in the proof of (12), we have $\widehat{\varphi}_t(x) = \sum_y p_{t|0}^r(x|y) \widehat{\varphi}_0(y)$, and hence by definition the

optimal $q$ to the backward half bridge problem is $q^{\widehat{\varphi}_1^{(k)}}$. On the other hand, by similar arguments as (22) and (25),

$$\frac{\widehat{\varphi}_1^{(k)}(z)}{\widehat{\varphi}_1^{(k)}(y)} = \mathbb{E}_{p_{t|1}^{(k)}(x|y)}\frac{p_{1|t}^r(z|x)}{p_{1|t}^r(y|x)} = \mathbb{E}_{p_{t|1}^{(k)}(x|y)}\frac{\widehat{\varphi}_0^{(k)}(x-y+z)}{\widehat{\varphi}_0^{(k)}(x)} = \mathbb{E}_{p_{t|1}^{(k)}(x|y)}\frac{(\mu/\varphi_0^{(k)})(x-y+z)}{(\mu/\varphi_0^{(k)})(x)}.$$

On the other hand, using the property of Bregman divergence, the unique fixed-point of (corr-DM) satisfies

$$\frac{\widehat{\varphi}_1^{(k)}(x_1^{d\leftarrow n})}{\widehat{\varphi}_1^{(k)}(x_1)} = \mathbb{E}_{p_{t|1}^{(k)}(x|x_1)}\frac{p_{1|t}^r(x_1^{d\leftarrow n}|x)}{p_{1|t}^r(x_1|x)},$$

while the unique fixed-point of (corr-AM) satisfies

$$\frac{\widehat{\varphi}_1^{(k)}(x_1^{d\leftarrow n})}{\widehat{\varphi}_1^{(k)}(x_1)} = \mathbb{E}_{p_{0|1}^{(k)}(x_0|x_1)}\frac{(\mu/\varphi_0^{(k)})(x_0^{d\leftarrow x_0^d+n-x_1^d})}{(\mu/\varphi_0^{(k)})(x_0)}.$$

By comparing the three equations above, the proof is complete. $\qquad\square$

### C.12. Other Omitted Results and Proofs

**Proposition 8.** *Assume $\mu = p_{\mathrm{unif}}$. Using the reference transition rate (16), initializing the controller to be one (i.e., let $\Phi_t^{(0)}(x)_{d,n} = 1$ for all $n \neq x^d$) and alternatively training the corrector and the controller is equivalent to initializing the corrector to be one (i.e., let $\widehat{\Phi}^{(0)}(x)_{d,n} = 1$ for all $n \neq x^d$) and alternatively training the controller and the corrector.*

*Proof.* Suppose we let $\Phi_t^{(0)}(x)_{d,n} = 1$ for all $n \neq x^d$. Then the optimal corrector can be computed as follows:

$$\widehat{\Phi}^{(1)}(x_1)_{d,n} = \mathbb{E}_{p_{t|1}^r(x|x_1)}\frac{p_{1|t}^r(x_1^{d\leftarrow n}|x)}{p_{1|t}^r(x_1|x)} = \sum_x \frac{p_{t|1}^r(x|x_1)}{p_{1|t}^r(x_1|x)}p_{1|t}^r(x_1^{d\leftarrow n}|x)$$

$$= \sum_x \frac{p_t^r(x)}{p_1^r(x_1)}p_{1|t}^r(x_1^{d\leftarrow n}|x) = \frac{1}{p_1^r(x_1)}\sum_x p_t^r(x)p_{1|t}^r(x_1^{d\leftarrow n}|x) = \frac{p_1^r(x_1^{d\leftarrow n})}{p_1^r(x_1)}.$$

Thus, under (16), when $p_0^r = \mu = p_{\mathrm{unif}}$, one has $p_1^r = p_{\mathrm{unif}}$, so $\widehat{\Phi}^{(1)}(x_1)_{d,n} = 1$ for all $n \neq x_1^d$. $\qquad\square$

**Remark 6.** *We remark that this choice coincides with the optimal corrector when assuming $p^r$ is memoryless, since under memoryless condition $\widehat{\varphi}_1 \propto p_1^r$.*

**Lemma 4.** *For a CTMC $(X_t)_{t\in[0,1]} \sim p^u$ with transition rate $u_t$ and any function $g : [0,1] \times \mathcal{X} \times \mathcal{X} \to \mathbb{R}$, we have*

$$\mathbb{E}_{p^u(X)}\sum_{t:X_{t_-}\neq X_t}g(t,X_{t_-},X_t) = \mathbb{E}_{p^u(X)}\int_0^1\sum_{y\neq X_t}g(t,X_t,y)u_t(y,X_t)\mathrm{d}t.$$

*Proof.* Consider the time-discretization: $\Delta t = \frac{1}{N}$ and $t_n = n\Delta t$. Then,

$$\mathbb{E}_{p^u(X)}\sum_{t:X_{t_-}\neq X_t}g(t,X_{t_-},X_t) = \mathbb{E}_{p^u(X)}\sum_{n=0}^{N-1}1_{X_{t_n}\neq X_{t_{n+1}}}g(t_n,X_{t_n},X_{t_{n+1}}) + O(\Delta t)$$

$$= \sum_{n=0}^{N-1}\sum_{x,y}p_{t_n}^u(x)p_{t_{n+1}|t_n}^u(y|x)1_{x\neq y}g(t_n,x,y) + O(\Delta t)$$

$$= \sum_{n=0}^{N-1}\sum_x p_{t_n}^u(x)\sum_{y\neq x}u_{t_n}(y,x)g(t_n,x,y)\Delta t + O(\Delta t)$$

$$= \mathbb{E}_{p^u(X)}\int_0^1\sum_{y\neq X_t}g(t,X_t,y)u_t(y,X_t)\mathrm{d}t.$$

$\qquad\square$

# D. Experimental Details and Additional Results

## D.1. Target Distributions

We consider Ising and Potts models on a square lattice $\Lambda = [L]^2$ with $L$ sites per dimension. We write $i \sim j$ if $i, j \in \Lambda$ are adjacent on the lattice. For simplicity, we impose periodic boundary conditions in both the horizontal and vertical directions. Both target distributions can be written in the form of

$$\nu(x) = \frac{1}{Z} e^{-\beta E(x)},$$

where $E$ is the energy function (Hamiltonian), $\beta > 0$ is the inverse temperature, and $Z = \sum_{x \in \mathcal{X}} e^{-\beta E(x)}$ is the partition function.

For Ising model with interaction parameter $J \in \mathbb{R}$ and external magnetic field $h \in \mathbb{R}$, the energy is

$$E_{\mathrm{Ising}}(x) = -J \sum_{i \sim j} x^i x^j - h \sum_i x^i, \ x \in \{\pm 1\}^\Lambda.$$

We keep $J = 1$ and $h = 0$ throughout the experiments. For Potts model with $N$ states and interaction parameter $J \in \mathbb{R}$, the energy is

$$E_{\mathrm{Potts}}(x) = -J \sum_{i \sim j} 1_{x^i = x^j}, \ x \in [N]^\Lambda.$$

We keep $J = 1$ throughout the experiments. Finally, we can compute the discrete score (first-order oracle) of these two distributions as follows:

$$\frac{\nu(x^{i \leftarrow n})}{\nu(x)} = \begin{cases} \exp\left(\beta(n - x^i)\left(J \sum_{j:\ j \sim i} x^j + h\right)\right), \ \forall n \in \{\pm 1\}, \quad \text{Ising}; \\ \exp\left(\beta J \left(\sum_{j:\ j \sim i} \left(1_{x^j = n} - 1_{x^j = x^i}\right)\right)\right), \ \forall n \in [N], \quad \text{Potts}. \end{cases} \tag{40}$$

## D.2. Implementation Details

**Model Backbone** We follow the implementation of MDNS (Zhu et al., 2025a) to use vision transformers (ViT, Dosovitskiy et al. (2021)) to serve as the backbone for the discrete diffusion model. In particular, we use the DeiT (Data-efficient image Transformers) framework (Touvron et al., 2021) with 2-dimensional rotary position embedding (Heo et al., 2025), which better captures the 2-dimensional spatial structure of the Ising and Potts models. While MDNS's model only requires the position input $x \in [N]^D$, in our learning objectives, the controller also receives time input $t \in [0, 1]$. Hence, we adopt the adaptive layer normalization (adaLN) mechanism in the DiT (Peebles & Xie, 2023) and SiT (Ma et al., 2024) to deal with the time conditioning. For learning $24 \times 24$ Ising model and $16 \times 16$ Potts model with 4 states, we use a model with 6 blocks, hidden dimension 32, and 4 heads. The total number of parameters in the model is around 144k (controller, with time conditioning) or 95k (corrector, without time conditioning).

**Training** Among all the training tasks, we use the AdamW optimizer (Loshchilov & Hutter, 2019) with a constant learning rate, exponential moving average (EMA) of model parameters with decay rate 0.9999, the generalized KL divergence[2] as the Bregman divergence, uniform time weight $w_t \equiv 1$, and the modified log-linear noise schedule $\gamma_t = \gamma/(t + \alpha)$ with $\alpha = 0.5$. Training proceeds in 5 outer alternating stages, each consisting of $T_{\mathrm{ctrl}}$ controller updates followed by $T_{\mathrm{corr}}$ corrector updates; we report the model at the end of stage 5. A buffer of size $|B|$ is maintained: every $T_r$ gradient updates, we use the EMA controller to draw fresh $(x_0, x_1)$ pairs via $\tau$-leaping with $L_\tau$ steps, push them into the buffer, and discard the oldest entries. Each gradient update samples a mini-batch of size $b$ (per GPU) from the buffer and re-evaluates $K$ values of $t \sim \mathrm{Unif}[0, 1]$ per pair. Per-temperature hyperparameters are listed in Tab. 4; all experiments use NVIDIA RTX A6000 GPUs.

Initial distribution $\mu$ and corrector loss differ across temperatures. For $\beta_{\mathrm{high}}$, we use uniform initialization $\mu = p_{\mathrm{unif}}$ and the AM loss (corr-AM) for the corrector; for $\beta_{\mathrm{critical}}, \beta_{\mathrm{low}}$, we observe that uniform initialization does not converge well, so we set $\mu$ to the zero-temperature distribution (i.e., $\mathrm{Unif}\{\pm \mathbf{1}\}$ for Ising and $\mathrm{Unif}\{\mathbf{1}, \mathbf{21}, ..., N\mathbf{1}\}$ for Potts) and use the DM loss (corr-DM) for the corrector since $\mu$ is then not positive everywhere.

The learning rate is also temperature-dependent. We default to $1e-3$ but reduce it to $5e-4$ for Ising $\beta_{\mathrm{critical}}, \beta_{\mathrm{low}}$ and Potts $\beta_{\mathrm{critical}}$, where preliminary experiments showed that $1e-3$ may lead to training instability.

*Table 4.* Per-temperature hyperparameters used in our experiments. "#GPU" denotes the number of GPUs used for training; the effective batch size per gradient step is $b \times$ #GPU. The † entries deviate from the default $\gamma = 1$: low-temperature runs use $\gamma = 0.25$, since the lower transition rate stabilizes training near the low-temperature ground state.

| Setting | blocks | #GPU | $b$ | $K$ | $\lvert B \rvert$ | $T_{\mathrm{ctrl}}$ | $T_{\mathrm{corr}}$ | $T_r$ | $L_\tau$ | lr |
|---|---|---|---|---|---|---|---|---|---|---|
| Ising $\beta_{\mathrm{high}} = 0.28$ | 6 | 2 | 64 | 8 | 512 | 400 | 200 | 20 | 200 | $1e-3$ |
| Ising $\beta_{\mathrm{critical}} = 0.4407$ | 6 | 2 | 64 | 8 | 512 | 400 | 200 | 20 | 200 | $5e-4$ |
| Ising $\beta_{\mathrm{low}} = 0.6$ † | 6 | 2 | 64 | 8 | 512 | 500 | 250 | 20 | 200 | $5e-4$ |
| Potts $\beta_{\mathrm{high}} = 0.9$ | 6 | 2 | 128 | 16 | 512 | 500 | 250 | 20 | 100 | $1e-3$ |
| Potts $\beta_{\mathrm{critical}} = 1.0986$ | 6 | 2 | 128 | 16 | 4096 | 200 | 100 | 10 | 100 | $5e-4$ |
| Potts $\beta_{\mathrm{low}} = 1.3$ † | 6 | 2 | 128 | 16 | 4096 | 500 | 250 | 20 | 100 | $1e-3$ |

**Generating Baseline and Ground Truth Samples**   We follow the implementation in existing literature (Guo et al., 2026) for baselines. For the learning-based baseline, we train LEAPS and MDNS on $24 \times 24$ Ising model and $16 \times 16$ Potts model for up to $150$k steps for each temperature. For MDNS, we apply the warm-up strategy (Zhu et al., 2025a) to initialize the training under $\beta_{\mathrm{critical}}$ from the pretrained checkpoint for $\beta_{\mathrm{high}}$, and initialize the training under $\beta_{\mathrm{low}}$ from the pretrained checkpoint for $\beta_{\mathrm{critical}}$. The Metropolis-Hastings (MH) and Swendsen-Wang (SW) sampling exactly follows the implementation in Guo et al. (2026) that ensures sufficient mixing.

**Evaluation**   We follow the procedure detailed in Zhu et al. (2025a, App. D.3, E.2) for the computation of **magnetization** and **2-point correlation error**. To compute the empirical **energy Wasserstein-2 distance** for two sets of samples $\{x_i\}$ and $\{y_j\}$, we obtain the energies of the datasets $\mathcal{E}_1 = \{E(x_i)\}$ and $\mathcal{E}_2 = \{E(y_j)\}$, and use the function `np.sqrt(ot.wasserstein_1d(`$\mathcal{E}_1$, $\mathcal{E}_2$, `p = 2))` from the POT package (Flamary et al., 2021).

**Effective Sample Size (ESS)**   We follow the practice in existing literature on neural samples (Zhang & Chen, 2022; Holderrieth et al., 2025; Zhu et al., 2025a). For a batch of i.i.d. samples $\{x_i\}_{i \in [B]}$ from $p$, suppose we associate each sample $x_i$ with a weight $w_i = \frac{\widehat{q}(x_i)}{p(x_i)}$, where $\widehat{q}$ is the unnormalized probability density / mass function of a probability distribution $q$, then the (normalized) **effective sample size (ESS)** of the samples with respect to $q$ is defined as $\frac{(\frac{1}{B} \sum_i w_i)^2}{\frac{1}{B} \sum_i w_i^2} \in \left[ \frac{1}{B}, 1 \right]$.

**Ablation Studies on TM v.s. DM (Fig. 1)**   We use a model with $4$ blocks, hidden dimension $32$ and $4$ heads. We use a batch size of $256$ and in (ctrl-DM) and (ctrl-AM), we sample $32$ $t$'s following $\mathrm{Unif}[0,1]$ for each pair of $(x_0, x_1)$. All three cases are trained under the same random seed for $5000$ steps, with a buffer size of $1024$ and resampling frequency $20$ with generation NFE $100$.

**Ablation Studies on Noise Schedules (Figs. 2 and 3)**   We use a model with $4$ blocks, hidden dimension $32$ and $4$ heads. We use a batch size of $64$ and in (ctrl-AM), we sample $8$ $t$'s following $\mathrm{Unif}[0,1]$ for each pair of $(x_0, x_1)$. All runs are trained under the same random seed for $5$ epochs with $200$ controller update steps and $200$ corrector update steps. The buffer size is $256$ and resampling frequency is $20$.

### D.3. Further Experimental Results

**Ablation Study: Noise Schedule**   We provide further ablation study for the constant noise schedule $\gamma_t \equiv \gamma$ in Fig. 3. A similar trend occurs with the modified log-linear schedule: the generated samples reach the best quality at around $\gamma \in [0.5, 1]$.

**Full Tables of Runtime**   We provide the full tables of runtime for the Ising and Potts models in Tabs. 5 and 6.

**Comparison with Analytical Solutions for Ising Models**   In this part, we compare the performance of DASBS with the analytical solutions for the lattice Ising model with periodic boundary conditions (Ferdinand & Fisher, 1969). We first introduce the method to estimate the normalizing constant of the target distribution $\nu(x) = \frac{1}{Z} \mathrm{e}^{-\beta E(x)}$ under the DASBS framework.

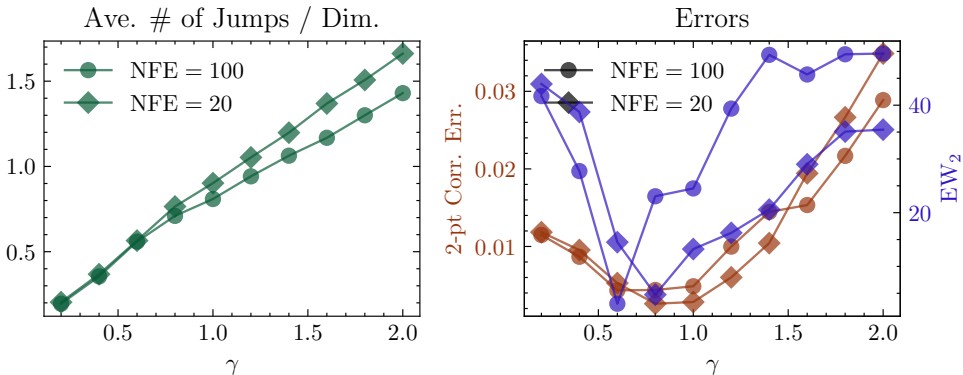

*Figure 3.* Ablation study of the hyperparameter $\gamma$ for the constant noise schedule $\gamma_t \equiv \gamma$ on Ising model with $L = 24$ and $\beta_{\text{high}} = 0.28$. NFE is the number of function evaluations during generation for both training and inference. *Left*: average number of jumps for each dimension during generation. *Right*: 2-point correlation error and energy Wasserstein-2 distance to ground-truth samples drawn from SW algorithm.

*Table 5.* Full table of runtime for learning to sample from the Ising model with $L = 24$. **Best** and second best results are highlighted. $*$: Wall-clock time measured on 2 NVIDIA RTX A6000 GPUs. $\dagger$: For $\beta_{\text{critical}}$ and $\beta_{\text{low}}$, using warm-up strategy in PDNS (Guo et al., 2026). $\ddagger$: Failed to converge to meaningful distributions at $\beta_{\text{critical}}$ and $\beta_{\text{low}}$ even with warm-up.

| Type | Inv. Temp. | $\beta_{\text{high}} = 0.28$ | | $\beta_{\text{critical}} = 0.4407$ | | $\beta_{\text{low}} = 0.6$ | |
|------|------------|------------------|-------------|------------------|-------------|------------------|-------------|
| | Metrics ↓ | Steps ($\times 1e3$)$^*$ | Runtime (h) | Steps ($\times 1e3$)$^*$ | Runtime (h) | Steps ($\times 1e3$)$^*$ | Runtime (h) |
| Uniform | DASBS | **3.0** | **0.2** | **3.0** | **0.2** | **3.75** | **0.3** |
| | LEAPS | 30 | 8.4 | 30 | 8.4 | 30 | 8.4 |
| | UDNS$^\dagger$ | 50 | 11.9 | - | - | - | - |
| | DFNS$^\ddagger$ | 50 | 2.1 | - | - | - | - |
| Masked | MDNS | 50 | 16.8 | 100 | 33.6 | 100 | 33.6 |

Let the *proposal path measure* be $p^u$ with learned controller. Define the *target path measure* as

$$p^t(X_{[0,1]}) = p^r(X_{[0,1)}|X_1)\nu(X_1) = \frac{1}{Z}p^r(X_{[0,1)})\frac{\mathrm{e}^{-\beta E(X_1)}}{p_1^r(X_1)} =: \frac{1}{Z}\widetilde{p}^t(X_{[0,1]}),$$

where $p^r$ is the reference path measure starting from $p_0^r = \mu$ with transition rate $r$ (16). Then an *unbiased* estimator of $Z$ is

$$Z = \mathbb{E}_{p^u(X_{[0,1]})}\frac{\widetilde{p}^t(X_{[0,1]})}{p^u(X_{[0,1]})} = \mathbb{E}_{p^u(X_{[0,1]})}\frac{p^r(X_{[0,1]})}{p^u(X_{[0,1]})}\frac{\mathrm{e}^{-\beta E(X_1)}}{p_1^r(X_1)}. \tag{41}$$

The **free energy** is defined as $\mathcal{F} = -\frac{1}{\beta}\log Z$. The **internal energy** is defined as $\mathcal{U} = \mathbb{E}_\nu E$ and directly estimated as the average energy of the samples. Finally, the **entropy** is defined as $\mathcal{S} = \beta(\mathcal{U} - \mathcal{F}) = -\mathbb{E}_\nu \log \nu$.

In (41), one requires to compute $p_1^r$. For the case where $\mu = \text{Unif}(\mathcal{X})$, it is obvious that $p_1^r = \mu$ is the invariant distribution of the CTMC with transition rate $r$ (16). For the case where $\mu$ is the zero-temperature distribution $\mu(x) = \frac{1}{N}\sum_{n=1}^N 1_{x=n\mathbf{1}}$, $p_1^r$ can be computed analytically:

$$p_1^r(x) = \sum_{y \in \mathcal{X}} p_{1|0}^r(x|y)\mu(y) = \sum_{n=1}^N p_{1|0}^r(x|n\mathbf{1})\frac{1}{N} = \frac{1}{N}\sum_{n=1}^N A(0,1)^{d_{\mathrm{H}}(x,n\mathbf{1})}B(0,1)^{D-d_{\mathrm{H}}(x,n\mathbf{1})},$$

where we used (31) to compute $p_{1|0}^r(x|n\mathbf{1})$.

*Table 6.* Full table of runtime for learning to sample from the Potts model with $L = 16$ and $N = 4$. **Best** and second best results are highlighted. ∗: Wall-clock time measured on 2 NVIDIA RTX A6000 GPUs.

| | Inv. Temp. | $\beta_{\text{high}} = 0.9$ | | $\beta_{\text{critical}} = 1.0986$ | | $\beta_{\text{low}} = 1.3$ | |
|---|---|---|---|---|---|---|---|
| Type | Metrics ↓ | Steps (×1e3)∗ | Runtime (h) | Steps (×1e3)∗ | Runtime (h) | Steps (×1e3)∗ | Runtime (h) |
| Uniform | DASBS | **3.75** | **0.3** | **1.5** | **0.2** | **3.75** | **0.3** |
| | LEAPS | 30 | 11.8 | 30 | 11.8 | 30 | 11.8 |
| Masked | MDNS | 50 | 17.6 | 100 | 35.2 | 100 | 35.2 |

We present the results in Tab. 7, where we report the mean and standard deviation of the three quantities via 32 independent runs, and each run uses 128 i.i.d. trajectories sampled from $p^u$ to provide one estimate of $(\mathcal{U}, \mathcal{S}, \mathcal{F})$. For DASBS and LEAPS, we use 100 NFE; for MDNS, we use $L^2 = 576$ NFE for precise computation of log RN derivatives in mask diffusion following the practice in MDNS. The DASBS numbers are obtained from the best-performing checkpoint among the three seeds. The results show that DASBS provides consistent estimates with theoretical values and is as good as or better than LEAPS and MDNS.

*Table 7.* Comparison with analytical solutions for the lattice Ising model with periodic boundary conditions for the case of $L = 24$ (mean ± standard deviation). The best result is highlighted in **bold**.

| | $\mathcal{U}/L^2$ | | | $\mathcal{S}/L^2$ | | | $\mathcal{F}/L^2$ | | |
|---|---|---|---|---|---|---|---|---|---|
| Inv. Temp. | $\beta_{\text{high}}$ | $\beta_{\text{critical}}$ | $\beta_{\text{low}}$ | $\beta_{\text{high}}$ | $\beta_{\text{critical}}$ | $\beta_{\text{low}}$ | $\beta_{\text{high}}$ | $\beta_{\text{critical}}$ | $\beta_{\text{low}}$ |
| Theoretical | $-0.6429$ | $-1.4402$ | $-1.9091$ | $0.5971$ | $0.2961$ | $0.0659$ | $-2.7753$ | $-2.1122$ | $-2.0189$ |
| DASBS | $-0.6538$ $\pm 0.0053$ | $-1.4367$ $\pm 0.0081$ | $\mathbf{-1.9087}$ $\pm 0.0031$ | $0.5899$ $\pm 0.0030$ | $\mathbf{0.2958}$ $\pm 0.0041$ | $\mathbf{0.0661}$ $\pm 0.0019$ | $-2.7607$ $\pm 0.0085$ | $\mathbf{-2.1079}$ $\pm 0.0027$ | $\mathbf{-2.0188}$ $\pm 0.0003$ |
| LEAPS | $\mathbf{-0.6426}$ $\pm 0.0060$ | $-1.2778$ $\pm 0.0093$ | $-1.6036$ $\pm 0.0034$ | $\mathbf{0.5963}$ $\pm 0.0017$ | $0.3655$ $\pm 0.0038$ | $0.2108$ $\pm 0.0036$ | $\mathbf{-2.7721}$ $\pm 0.0002$ | $-2.1085$ $\pm 0.0019$ | $-1.9550$ $\pm 0.0057$ |
| MDNS | $-0.6404$ $\pm 0.0057$ | $\mathbf{-1.4408}$ $\pm 0.0085$ | $-1.9053$ $\pm 0.0036$ | $0.6018$ $\pm 0.0024$ | $0.2998$ $\pm 0.0037$ | $0.0722$ $\pm 0.0028$ | $-2.7896$ $\pm 0.0052$ | $-2.1211$ $\pm 0.0033$ | $-2.0256$ $\pm 0.0024$ |

