# OpenReview forum: "Discrete Adjoint Schrödinger Bridge Sampler"
_ICML.cc/2026/Conference — ICML 2026 regular_

### Official Review · Reviewer_fjD9 · 2026-03-04

**Soundness:** 3
**Presentation:** 3
**Significance:** 3
**Originality:** 3
**Overall Recommendation:** 5
**Confidence:** 3

**Summary:**

This paper introduces **Discrete Adjoint Schrödinger Bridge Samplers**, an extension of the Adjoint Schrödinger Bridge Sampler to discrete state spaces. The primary objective is to **learn to sample from unnormalized target distributions**. The authors conduct experiments on the **Ising Model** and **Potts Model** at three distinct temperatures: **critical temperature, below, and above**. Performance is validated by comparing **2-point correlation, magnetization, and energy Wasserstein distance** against simulations using the Swendsen-Wang algorithm. Empirical results demonstrate that the proposed method **outperforms previous approaches in most cases**.

**Compliance With Llm Reviewing Policy:**

Affirmed.

**Final Justification:**

The paper demonstrates strong mathematical novelty and is evaluated on a sufficient range of benchmarks.

I initially had some questions about the writing and certain explanations. In addition, I requested a comparison to theoretically available exact values on the Isign Model, which the authors have now included.

Overall, I find the paper strong in terms of soundness, originality, significance, and clarity, and I therefore assign it a score of 5.

**Key Questions For Authors:**

## Questions for the Authors
1. **Periodic Boundary Conditions**: Are periodic boundary conditions used for the Ising model? If so, why was the comparison to theoretically available solutions (e.g., [1]) omitted?
2. **Weight w(t)**: What is the rationale behind introducing **w(t)**, and why is it set to 1 in practice? **How would alternative values affect the upper bound property of the joint KL with respect to the KL between marginals** ([6])?
3. **Trajectory Importance Weight**: How does **Theorem 5.2** justify omitting the trajectory importance weight? Could this be elaborated in more detail?
4. **Runtimes**: Why are runtimes only reported for high temperatures? **Are runtimes at the critical temperature significantly longer?**
5. **Stages vs. Epochs**: How are **"stages"** defined in the context of the algorithm? **Is this synonymous with "epochs"?**

## References
[1] Ferdinand, A. E., & Fisher, M. E. (1969). *Bounded and inhomogeneous Ising models. I. Specific-heat anomaly of a finite lattice.* Physical Review, 185(2), 832.

[2] Wu, D., Wang, L., & Zhang, P. (2019). *Solving statistical mechanics using variational autoregressive networks.* Physical Review Letters, 122(8), 080602.

[3] Nicoli, K. A., et al. (2020). *Asymptotically unbiased estimation of physical observables with neural samplers.* Physical Review E, 101(2), 023304.

[4] Sanokowski, Sebastian, et al. "Scalable Discrete Diffusion Samplers: Combinatorial Optimization and Statistical Physics." The Thirteenth International Conference on Learning Representations.

[5] Ou, Zijing, Ruixiang ZHANG, and Yingzhen Li. "Discrete Neural Flow Samplers with Locally Equivariant Transformer." The Thirty-ninth Annual Conference on Neural Information Processing Systems.

[6] Song, Y., et al. (2021). *Maximum likelihood training of score-based diffusion models.* Advances in Neural Information Processing Systems, 34, 1415–1428.

**Strengths And Weaknesses:**

## Strengths
- **Novelty**: The extension to discrete state spaces is a **significant and meaningful** contribution.
- **Performance**: The model **consistently outperforms** benchmark algorithms in empirical comparisons.
- **Validation**: Rigorous empirical validation and ablation studies **substantiate the claims**.


## Weaknesses & Suggestions

### Clarity and Justification
- **Line 380**: The use of **periodic boundary conditions** for the Ising model is unclear. If employed, the authors should explicitly state this and **consider comparing their results to theoretically available solutions** for free energy, internal energy, and entropy (e.g., [1], as in [3, 4, 5]).
- **Line 284**: The introduction of the weight **w(t)** lacks justification. While it is set to 1 in practice, the implications of alternative values—**particularly whether the algorithm still satisfies the upper bound property of the joint KL with respect to the KL between marginals** (as in [5])—should be discussed.
- **Theorem 5.2**: The connection between **Theorem 5.2** and the omission of the **trajectory importance weight** is unclear. A more detailed discussion would **strengthen the theoretical grounding**.

### Technical and Editorial Issues
- **Line 76**: The sentence appears **grammatically incomplete or broken**.
- **Line 359**: Runtimes are only reported for high temperatures. **Are runtimes at the critical temperature significantly longer?** This should be clarified.
- **Line 260**: The definition of **"stages"** is ambiguous. **Is this equivalent to what is typically referred to as an "epoch"?**

### Literature Review
- The discussion of **previous work on neural samplers** is incomplete. For example:
  - [2, 3] are seminal works in discrete neural sampling and employed **autoregressive neural samplers** for the Ising model.
  - [4] are the first to use **diffusion samplers** on the Ising model, but is only cited in the context of **combinatorial optimization (CO)** in this paper.

---

> ### Author Rebuttal · Authors · 2026-03-31
>
> We thank the reviewer for the instructive feedback. We are glad that DASBS is perceived as a significant and meaningful contribution.
> # 1. Periodic boundary conditions and comparison with analytical solutions
> We confirm that periodic boundary condition is used following the standard practice, which has been stated in line 1341 and will be highlighted in main text in revision.
>
> Following the reviewer's suggestion, we have computed the requested quantities and evaluated their empirical estimates. The table below shows the comparison of per-spin internal energy ${\cal U}/L^2$ on $2^{12}$ samples, validating that DASBS can generate accurate samples:
>
> |$\beta$|High|Critical|Low|
> |:-:|:-:|:-:|:-:|
> |Theoretical|-0.6429|-1.4402|-1.9091|
> |DASBS|-0.6566|-1.4408|-1.9086|
> |LEAPS|-0.6389|-1.2752|-1.6031|
> |UDNS|-0.6039|-|-|
> |DFNS|-1.7908|-|-|
> |MDNS|-0.6419|-1.4397|-1.9011|
> |MH|-0.6431|-1.4404|-1.9093|
>
> Next, we estimate the per-spin entropy ${\cal S}/L^2$ and free energy ${\cal F}/L^2$ under the same setting:
>
> |||${\cal U}/L^2$|||${\cal S}/L^2$|||${\cal F}/L^2$||
> |:-:|:-:|:-:|:-:|:-:|:-:|:-:|:-:|:-:|:-:|
> |$\beta$|High|Critical|Low|High|Critical|Low|High|Critical|Low|
> |Theoretical|-0.6429|-1.4402|-1.9091|0.5971|0.2961|0.0659|-2.7753|-2.1122|-2.0189|
> |DASBS|-0.6566|-1.4408|-1.9086|0.6650|0.4335|0.5058|-3.0315|-1.4408|-2.7516|
>
> The relatively large gap between the theoretical and empirical estimates is **an estimation artifact rather than a sampling failure**, as we approximate $\log p^u_1(x_1)$ using ELBO with the variational proposal being the reference process $p^r$:
>
> $$\log p^u_1(x_1)\ge\log p^r_1(x_1)+\mathbb{E} _ {p^r(x_{t_0,...,t_{N-1}}|x_1)}\log\frac{dp^u}{dp^r}(x_{t_0,...,t_N}),$$
> where $t_n=n/N,~N=200$.
>
> This loose bound stems from a fundamental methodological difference. Methods like DFNS [5] explicitly train bidirectional CTMC dynamics to close the Jarzynski gap, and discrete diffusion samplers [4], like standard diffusion models, can naturally use the exact noising process as the proposal. In contrast, our SB framework focuses exclusively on learning a forward sampling process $p^u$ without explicitly learning the backward transition (which requires $\frac{\hat\varphi_t(x^{d\gets n})}{\hat\varphi_t(x)}$ for *all* $t$) needed to form a tight backward proposal. Thus, DASBS is not designed to be a tight likelihood estimator. Extending our framework to co-train the backward dynamics for tight free energy estimation is an exciting avenue for future work.
>
> # 2. Weight $w_t$
>
> We thank the reviewer for raising this insightful question. Setting $w_t$ to specific values (e.g., $\gamma_t/N$ in our case, analogous to $\lambda(t)=g(t)^2$ in [6]) makes the loss strictly equivalent to the trajectory KL divergence, thereby establishing an upper bound on the marginal KL. However, departing from this choice does **not** compromise the theoretical rigor:
>
> 1. The matching relations (19, 22, 25) are decoupled across time $t$. As long as $w_t>0$, the global optimal solution remains unchanged.
> 2. Similar to the empirical observations in [6], strict likelihood weighting often assigns large weights to time with small noise scales, causing instability. Employing a uniform weight (a standard practice widely adopted in flow matching) ensures stable targets across all times.
>
> # 3. Theorem 5.2, trajectory importance weight
>
> Assuming exact corrector, the trajectory importance weight $p^\star/p^u$ theoretically enables *single-step optimization* by correcting the expectation from current policy $p^u$ to $p^\star$, as the variational expressions (20, 23) are under $p^\star$. However, estimating $p^\star/p^u$ introduces severe variance and error due to discretization and approximation (see remarks after Lem. B.7). Consequently, dropping it strictly benefits empirical convergence (Fig. 1).
>
> Thm. 5.2 rigorously justifies this omission by proving that the unweighted alternating process still converges to $p^\star$. We show that the optimal unweighted updates correspond to **solving half-bridges**, and thus the resulting update is alternatingly projecting the current path measure onto the two constraint sets defined by the optimal controller and corrector our alternating scheme, which is exactly the **IPF** procedure. We will clarify this in revision.
> # 4. Line 76
> We thank the reviewer for pointing out this issue and will correct it in revision.
> # 5. Runtimes
> Due to space limit, please refer to point 3 in our response to reviewer 1dhf where full runtime tables are detailed.
> # 6. Stages v.s. epochs
> Yes, they are synonymous in our context. Each stage/epoch refers to one full cycle of sequentially optimizing the controller (500 steps) and the corrector (250 steps). We will clarify this in revision.
> # 7. Missing literature
> We appreciate this suggestion and will add them in revision.
>
> We thank the reviewer again for the professional review and hope that our responses have addressed the concerns. We are happy to provide further clarifications if needed.

---

> > ### Author Rebuttal · Reviewer_fjD9 · 2026-04-01
> >
> > We thank the authors for their detailed response. Most of our questions have been satisfactorily addressed. However, I still believe that the experimental evaluation of free-energy and entropy values, due to the usage of the ELBO, is not yet fully complete.
> >
> > In particular, estimating $\log Z$ does not require relying on lower bounds. Let $\tilde{p}(x)$ denote the unnormalized target density and let $q_\theta(x)$ be the sampler distribution. Then $Z=\int \tilde{p}(x) dx =
> > E_{x \sim q_\theta} [\frac{\tilde{p}(x)}{q_\theta(x)}],$
> > which yields the standard importance-sampling estimator
> > $
> > \hat Z=\frac{1}{N}\sum_{i=1}^N \frac{\tilde{p}(x_i)}{q_\theta(x_i)},
> >  x_i\sim q_\theta.
> > $
> > Accordingly, $\log Z$ can be estimated directly from samples, without introducing an ELBO-based lower bound. In practice, this can be computed stably via a log-sum-exp implementation.
> > We believe this would provide a more complete and informative evaluation of the reported free-energy quantities.
> >
> > Moreover, this idea can be extended naturally to diffusion-based samplers by replacing the density ratio above with the ratio between the forward and reverse processes. For a sufficient number of samples, this estimator has been shown to converge to the theoretical free energy values if the sampler does not suffer from mode collapse.

---

> > > ### Author Response · Authors · 2026-04-01
> > >
> > > We sincerely thank the reviewer again for the insightful comments and for pointing out a better way of free energy estimation without the ELBO approximation. We are glad to hear that most of the raised concerns have been addressed in our rebuttal.
> > >
> > > As per the reviewer's suggestion, we leverage the following method to estimate the normalizing constant of the target distribution $\nu(x)=\frac1Z\mathrm{e}^{-\beta E(x)}$.
> > >
> > > Let the *proposal path measure* be $p^u$ with learned controller (line 290).
> > > Define the *target path measure* as
> > > $$
> > > p^t(X _ {[0,1]})=p^r(X _ {[0,1)}|X _ 1)\nu(X _ 1)=\frac1Zp^r(X _ {[0,1]})\frac{\mathrm{e}^{-\beta E(X _ 1)}}{p^r _ 1(X _ 1)}=:\frac1Z\widetilde{p}^t(X _ {[0,1]}),
> > > $$
> > > where $p^r$ is the reference path measure starting from $p^u _ 0=\mu$ with transition rate $r$ (16). Then an **unbiased** estimator of $Z$ is
> > > $$
> > > Z=\mathbb{E} _ {p^u(X _ {[0,1]})}\frac{\widetilde{p}^t(X _ {[0,1]})}{p^u(X _ {[0,1]})}=\mathbb{E} _ {p^u(X _ {[0,1]})}\frac{p^r(X _ {[0,1]})}{p^u(X _ {[0,1]})}\frac{\mathrm{e}^{-\beta E(X _ 1)}}{p^r _ 1(X _ 1)}.
> > > $$
> > >
> > > The free energy is ${\cal F}=-\frac1\beta\log Z$. The entropy is then estimated via the relation ${\cal S}=\beta({\cal U-F})$.
> > >
> > > We present the updated results in the following table, where we report the mean and standard deviation of the three quantities via $32$ independent runs, and each run uses $128$ i.i.d. trajectories sampled from $p^u$ to provide one estimate of $({\cal U,S,F})$. For DASBS and LEAPS, we use $100$ NFE; for MDNS, we use $L^2=576$ NFE for precise computation of log RN derivatives in mask diffusion following the practice in MDNS.
> > >
> > > |||${\cal U}/L^2$|||${\cal S}/L^2$|||${\cal F}/L^2$||
> > > |:-:|:-:|:-:|:-:|:-:|:-:|:-:|:-:|:-:|:-:
> > > |$\beta$|High|Critical|Low|High|Critical|Low|High|Critical|Low|
> > > |Theoretical|-0.6429|-1.4402|-1.9091|0.5971|0.2961|0.0659|-2.7753|-2.1122|-2.0189|
> > > |DASBS|-0.6538 ± 0.0053|-1.4367 ± 0.0081|**-1.9087** ± 0.0031|0.5899 ± 0.0030|**0.2958** ± 0.0041|**0.0661** ± 0.0019|-2.7607 ± 0.0085|**-2.1079** ± 0.0027|**-2.0188** ± 0.0003
> > > |LEAPS|**-0.6426** ± 0.0060|-1.2778 ± 0.0093|-1.6036 ± 0.0034|**0.5963** ± 0.0017|0.3655 ± 0.0038|0.2108 ± 0.0036|**-2.7721** ± 0.0002|-2.1085 ± 0.0019|-1.9550 ± 0.0057
> > > |MDNS|-0.6404 ± 0.0057|**-1.4408** ± 0.0085|-1.9053 ± 0.0036|0.6018 ± 0.0024|0.2998 ± 0.0037|0.0722 ± 0.0028|-2.7896 ± 0.0052|-2.1211 ± 0.0033|-2.0256 ± 0.0024
> > >
> > > In the table, the best results are in bold. The results show that DASBS provides consistent estimates with theoretical values and is comparable and even better than LEAPS and MDNS. We will include this result in the appendix.
> > >
> > > We thank the reviewer again for the professional and constructive comments. According to the ICML policy, this is our final response. If it has fully addressed the concerns, we would appreciate it if the reviewer could adjust the final score accordingly to reflect the improvements.

---

### Official Review · Reviewer_mMjw · 2026-03-08

**Soundness:** 3
**Presentation:** 3
**Significance:** 3
**Originality:** 3
**Overall Recommendation:** 5
**Confidence:** 4

**Summary:**

This paper provides an extension of the SB and SOC frameworks to discrete state spaces. Based on this theoretical formulation, the paper proposes a discrete version of adjoint matching and denoising matching under the assumption of a cyclic group structure on the discrete space necessary to enable additive noise transitions. They then compare the proposed discrete samplers, include connections with other works and provide convergence guarantees. Finally, there is empirical evidence of the method on the potss and ising models.

**Compliance With Llm Reviewing Policy:**

Affirmed.

**Final Justification:**

The rebuttal cleared my concerns, especially regarding the cyclic group assumption. I will keep my positive score

**Key Questions For Authors:**

1. In the experiments section there is a comparison about the training efficiency in terms of run time, could you also provide how do the different methods compare in terms of energy evaluations?

2. Can you comment or provide some analysis in relationship with the scalability of the method as the number of categories and dimensions increase?

3. Im a bit concerned by the very low ESS shown in figure 1. I read in line 1115 the remark where you provide some comments about the possible causes. How can you reliably trust the output with such a low ESS, as there might be high autocorrelation between samples. Could you provide further insights on this and how this can be mitigated?

**Limitations:**

yes

**Strengths And Weaknesses:**

Strengths:
- Presentation: table 1 really helps to follow the paper. The overall presentation is good.
- Soundness: the method proposed method has a strong theoretical justification . The extension of SB and SOC to CTMC is very elegant. I also think that the use of cyclic group structure for additive noise is very clever.
- Significance: I understand that the theoretical analysis in discrete domains is very interesting, due to the cyclic group assumption the significance is a bit limited. (see more in weaknesses)
- Originality: the work provides new insights on discrete samplers. Extending continuous frameworks to the discrete domains and highlighting the need for the cyclic group structure. They also propose an efficient sampler for this case which has benefits over previous works under the cyclic group assumption.

Weaknesses:
- As mentioned above due to the cyclic group assumption limits a lot the applicability of the method to lattice like and categorical product spaces. Many interesting discrete spaces like molecular structures or graphs lack this property.
- The controller matrix is of size DxN which can scale badly with dimensions and number of categories
- The convergence guarantee relies on classical IPF theory, which assumes exact optimization of each alternating step. However, in practice these steps are implemented via neural networks.

---

> ### Author Rebuttal · Authors · 2026-03-31
>
> We thank the reviewer for the thoughtful review and constructive feedback. We are glad that the reviewer likes the presentation and finds our theoretical framework elegant and insightful.
>
> # 1. Cyclic group assumption and applicability
>
> Due to space limit, please refer to point 2 in our response to reviewer HN4f. In short, the cyclic group structure does not limit the applicability of our framework to any discrete product space, and applying our method to complex structures like molecular graphs is a promising future direction.
>
> # 2. Controller matrix scalability
>
> We respectfully disagree with the claim about the scalability of the controller matrix, as our complexity is the same as typical discrete diffusion models [1]. For modeling a distribution on $[N]^D$, a discrete diffusion model inputs a state $x_t\in[N]^D$ and time $t$, and outputs $s_\theta(x_t,t)\in\mathbb{R}^{D\times N}$, where the $(d,n)$-th entry approximates $\frac{p_t(x^{d\gets n})}{p_t(x)}$, and $p_t$ is the marginal distribution of the forward (noising) process at time $t$. Our method has the same output dimension and similar computational complexity compared to standard discrete diffusion models.
>
> [1] Lou et al. Discrete Diffusion Language Modeling by Estimating the Ratios of the Data Distribution.
>
> # 3. Convergence guarantees with neural approximations
>
> We acknowledge the theoretical gap between the exact IPF optimization assumption and the practical use of neural network approximations. However, this is a standard and accepted gap shared across the neural Schrödinger bridge literature (e.g., [2, 3]). Empirically, modern neural networks possess sufficient capacity to approximate the optimal potentials, driving the alternating process to a high-quality equilibrium despite the lack of exact optimization at each step.
>
> [2] Shi et al. Diffusion Schrödinger Bridge Matching. NeurIPS 2023.
>
> [3] Liu et al. Adjoint Schrödinger Bridge Sampler. NeurIPS 2025.
>
> # 4. Energy evaluations and scalability
>
> Due to space limit, please refer to point 1 in our response to reviewer 1dhf. In short, for Ising/Potts models, we do not perform any energy evaluations since the entire 1st-order oracle can be computed analytically and parallelized via a single GPU operation. DASBS also requires significantly fewer oracle queries compared to the 0th-order baselines.
>
> # 5. Scalability with dimensions and categories
>
> We analyze the scalability of DASBS from two perspectives.
>
> ## (1) Network architecture scalability
>
> Due to our per-dimension independent transition parameterization, the two networks only need to output a transition rate matrix of size $D\times N$. In terms of neural network architecture, for modern architectures like the ViT used in our experiments, the true computational bottleneck is the self-attention, which scales as $O(D^2)$, while the $O(D\times N)$ output head is extremely lightweight. The parameterization of DASBS thus scales exactly as any standard discrete diffusion model. Empirically, our evaluation firmly supports this: we benchmarked DASBS on Ising ($D=576,N=2$) and Potts models ($D=256,N=4$). To the best of our knowledge, these represent the highest-dimensional state spaces currently evaluated in the discrete diffusion neural sampler literature.
>
> ## (2) Training oracle scalability
>
> As $D$ and $N$ increase, the primary scalability constraint lies in the 1st-order oracle $\left(\frac{\nu(x^{d\gets n})}{\nu(x)}\right)_{d,n\ne x^d}$. For target distributions with analytically tractable or localized energy functions (e.g., Ising/Potts), this can be parallelized with negligible overhead. However, we explicitly acknowledge that if the target energy is a computationally expensive, non-parallelizable black-box simulator (e.g., exact quantum physics simulations or complex fluid dynamics), evaluating the 1st-order oracle could indeed become a computational bottleneck.
>
> # 6. Low ESS and sample quality
>
> We thank the reviewer for carefully reading the paper and raising this thoughtful question. The ESS reported here is computed from the trajectory RN derivative $\frac{{\rm d}p^\star}{{\rm d}p^u}(X_{[0,1]})$. As explained in the remark after Lem. B.7, in uniform discrete diffusion, our estimator of the trajectory RN derivative typically exhibits high variance due to more frequent transitions and $\tau$-leaping discretization; also, we have an approximation error from the neural network (see the last step in the proof of Lem. B.7). All these reasons cause the ESS to remain small even when the marginal distributions have converged. Therefore, we do not rely on this trajectory importance reweighting for training. Moreover, our sample quality is reliably validated by other physical metrics (magnetization, 2-point correlation, EW2), which show good convergence to the target distribution.
>
> We thank the reviewer again for the insightful review and hope that our responses have addressed the concerns.

---

> > ### Author Rebuttal · Reviewer_mMjw · 2026-04-03
> >
> > Thanks for clarifying my questions

---

### Official Review · Reviewer_HN4f · 2026-03-10

**Soundness:** 4
**Presentation:** 3
**Significance:** 3
**Originality:** 4
**Overall Recommendation:** 5
**Confidence:** 3

**Summary:**

This paper presents a new framework for discrete Schroedinger bridge (SB) and stochastic optimal control (SOC).
Unlike the continuous case, the discrete case is non-trivial due to the lack of gradients and computational complexity.
The authors first show the relationship between SB and SOC in the discrete setting, where the optimal transition rate is free from the gradient.
To develop a discrete version of the adjoint Schroedinger bridge sampler (ASBS), they introduce discrete (controller and corrector) adjoint matching, which does not require gradient calculation.
Finally, a new sampler, discrete ASBS (DASBS), is proposed, based on an alternating fixed-point algorithm.

**Compliance With Llm Reviewing Policy:**

Affirmed.

**Final Justification:**

The authors fully addressed my concerns.

**Key Questions For Authors:**

+ The authors use a cyclic group structure. But the question is how universal this assumption is. In this study, the state space $\mathcal{X}$ looks too simple, but for more practical cases, is it also possible to use the same structure?

**Limitations:**

yes

**Strengths And Weaknesses:**

# Strength
+ The literature is well written.
+ After describing the difficulties in the discrete setting, the challenges are addressed based on a novel idea and theoretical analysis.
  - In Theorem 3.1, a discrete analogue of the optimal transition rate is established.
  - In the continuous case, the adjoint matching requires the gradient $\nabla \phi (x)$; it is replaced with the quantity of the form $\phi (y) / \phi(x)$ in the discrete case.
  - An alternating fixed-point algorithm is proposed, which appears efficient.
# Weaknesses
+ As the authors mentioned in the final section, there are several limitations. For example, the evaluation is currently limited to synthetic benchmarks.

---

> ### Author Rebuttal · Authors · 2026-03-31
>
> We thank the reviewer for the positive and constructive review. We are glad that the reviewer found our theoretical framework novel and the paper well-written.
>
> # 1. Limited evaluation
>
> We acknowledge that our current evaluation is limited to synthetic benchmarks and have explicitly acknowledged this as a limitation in Sec. 7. Lattice Ising and Potts models are the standard, rigorous testbeds in the discrete neural sampler and statistical physics literature, which exhibit phase transitions that effectively benchmark algorithmic properties. As our main contribution is methodological, these tasks provide the clearest and most standard validation of the discrete AM mechanics.
>
> # 2. Cyclic group structure, more general state spaces
>
> We thank the reviewer for this insightful question, which is a shared concern across multiple reviewers. We will jointly address all related questions on the cyclic group assumption in two parts:
>
> ## (1) On the necessity and universality of the cyclic group structure
>
> In the continuous AM framework, the additive nature of the reference noise (i.e., $p^r_{1|t}(y|x)=q_t(y-x)$) is fundamental to deriving the objectives. To seamlessly extend this to discrete domains, we mathematically require the reference transition kernel to exhibit shift invariance, which is critical for the derivations of (19, 22). However, a standard discrete space like $[N]^D$ is *not algebraically closed under addition*; simple shifting operations lead to out-of-bounds errors. Equipping it with a cyclic group structure provides the essential **algebraic closure** to formulate additive noise.
>
> On the other hand, it is crucial to emphasize that this cyclic group mapping is entirely **decoupled from the target distribution and the intrinsic semantics of the discrete state space**. It does not imply that the data must physically possess a cyclic topology or any ordinal relationship. For example, the states $[N]$ can represent purely categorical data without any inherent order (e.g., node types or vocabulary), or they can represent ordinal data (e.g., molecule counts). In both scenarios, mapping the states to $\mathbb{Z}_N$ simply allows us to perform the group operations required by our solver. Therefore, our framework is universally applicable.
>
> ## (2) On extension to more general state spaces
>
> Although we focus on ${\cal X}=[N]^D$, the framework can be generalized to **any discrete product space** ${\cal X}=\prod_{d=1}^DX_d$, where each $X_d$ is a finite set. Without loss of generality, we can always assume $X_d$ to be $[N_d]$. By treating ${\cal X}$ as $\prod_{d=1}^D\mathbb{Z}_{N_d}$, the same $r_t(y,x)$ (16) leads to a similar additive noise (17), and the rest of the AM mechanics follow with minor modifications to account for the different $N_d$ states across dimensions.
>
> Such a space can formally represent complex structures like **molecular graphs**, where ${\cal X}={\cal A}^V\times{\cal B}^{V(V-1)/2}$ (with ${\cal A}$ as atom types, ${\cal B}$ as bond types, and $V$ as the maximum number of atoms), and our framework is theoretically directly applicable. Integrating our core AM framework with advanced geometric architectures (e.g., equivariant GNNs) and designing constrained sampling mechanisms to handle chemical valency remains a non-trivial and highly promising direction for future work.
>
> We thank the reviewer again for the supportive review and insightful question regarding the cyclic group structure, and we hope that our response has clarified the universality of our framework and its potential extensions.

---

> > ### Author Rebuttal · Reviewer_HN4f · 2026-04-03
> >
> > I would like to thank the authors for addressing my concerns and questions.

---

### Official Review · Reviewer_1dhf · 2026-03-13

**Soundness:** 3
**Presentation:** 2
**Significance:** 3
**Originality:** 3
**Overall Recommendation:** 5
**Confidence:** 3

**Summary:**

The authors consider the problem of sampling from discrete probability distributions. They propose to generalize the adjoint matching (and the adjoint Schrödinger Bridge sampler) proposed previously to the continuous distribution in the case of discrete ones. Specifically, consider the Schrödinger Bridge problem for CTMC and highlight the principal problem of Adjoint Matching in discrete space: choosing the reference process in a way it’s transition kernel is additive. They solve this problem by considering the cyclic group structure for the discrete state space. Next the propose a new algorithm consisting of alternation between controller AM and corrector DM (corrector AM) steps and proof convergese for the (controller AM, corrector DM) part.

**Compliance With Llm Reviewing Policy:**

Affirmed.

**Final Justification:**

The authors fully addressed my concerns.

**Key Questions For Authors:**

- Is considering a cyclic structure the only way of extending AM from the continuous domain to the discrete domain?

**Limitations:**

yes

**Strengths And Weaknesses:**

Strengths:
- Generalization of adjoint matching on discrete domains is a strong theoretical contribution.
- The authors propose a new algorithm with convergence guarantees.
- The proposed algorithm shows comparable performance with other baselines while having faster training in some setups (Ising with \beta_{\text{high}}).
- The relation between DM and AM in continuous and discrete settings is explained very well.

Weaknesses:
- For a sequence of length D and N states, one needs to compute O(D*(N-1)) energy differences, which is a sufficient burden for scalability. In the current experimental setups, this problem is not striking since there are only N=2 or N=4 states. So the shown performance may be a great fit for this particular setup. Also, it is not clear whether the benefit comes from the proposed algorithm or from more evaluations of energy compared to other baselines.
- The absence of the training algorithm in the paper limits the clarity of the proposed method. Also, it is not clear from the main text which corrector is used in experiments, DM or AM.
- The runtime report is not complete. The authors report a runtime comparison for Ising models, but not for Potts models.
- On Potts models with higher \beta, the masked MDNS sampler still performs significantly better.
- The lack of convergence guarantees from (ctrl-AM, corr-AM), which is important since, for some experimental settings, authors also use this pair.

---

> ### Author Rebuttal · Authors · 2026-03-31
>
> We thank the reviewer for the detailed and constructive feedback.
>
> # 1. Energy evaluations and scalability
>
> This is a shared concern that has been acknowledged as a potential limitation in Sec. 7. We agree that for *complex, non-parallelizable* energy functions, computing $O(DN)$ energies sequentially would be a significant bottleneck. However, assuming our method requires $O(DN)$ times more evaluations misrepresents the computational reality of our setup for two reasons:
>
> 1. Highly vectorized 1st-order oracle: For Ising/Potts models, we **do not** perform any energy evaluations. The entire matrix of 1st-order oracle can be derived analytically (line 1362) and computed simultaneously via a single GPU operation. Thus, the actual wall-clock time to compute this structurally rich 1st-order oracle is comparable to evaluating the 0th-order oracle required by the baselines such as MDNS/UDNS/LEAPS/DFNS.
> 2. Number of oracle queries: Since the wall-clock cost per oracle query is comparable, the overall efficiency heavily favors DASBS. Based on our training configurations, for high temperature settings, DASBS requires only around 32k queries (5 stages × 25 resampling steps per stage × 256 buffer size), while the 0th-order baselines require at least 600k queries (e.g., 5000 epochs × 128 buffer size for MDNS, 3000 epochs × 200 buffer size for LEAPS). For critical and low temperature settings, DASBS/LEAPS can be trained from scratch with a similar number of queries, while MDNS requires warm start and thus the total number of queries is doubled, and UDNS/DFNS fail to converge even with warm start.
>
> # 2. Absent information
>
> We will include the full pseudocode in revision. The corrector loss used in our experiments has been detailed in App. C.2 under the "training" paragraph (lines 1377-1387). We will clarify this in the main text as well.
>
> # 3. Runtime comparison
>
> We present in the following two tables the full runtime for Ising and Potts models, respectively, and will include them in revision. For critical and low temperatures, DASBS and LEAPS are trained from scratch using the same number of steps, while MDNS requires warm-up from high-temperature checkpoints and thus the reported runtimes are doubled to account for the warm-up. UDNS and DFNS failed to converge to reasonable performance even with warm start.
>
> ||$\beta$|High|High|Critical|Critical|Low|Low|
> |:-:|:-:|:-:|:-:|:-:|:-:|:-:|:-:|
> |Type|Metrics↓|Steps (×1000)|Runtime (h)|Steps (×1000)|Runtime (h)|Steps (×1000)|Runtime (h)|
> |Uniform|DASBS|3.75|0.5|3.75|0.5|3.75|0.5|
> |Uniform|LEAPS|30|8.4|30|8.4|30|8.4|
> |Uniform|UDNS|50|11.9|-|-|-|-|
> |Uniform|DFNS|50|2.1|-|-|-|-|
> |Masked|MDNS|50|16.8|100|33.6|100|33.6|
>
> ||$\beta$|High|High|Critical|Critical|Low|Low|
> |:-:|:-:|:-:|:-:|:-:|:-:|:-:|:-:|
> |Type|Metrics↓|Steps (×1000)|Runtime (h)|Steps (×1000)|Runtime (h)|Steps (×1000)|Runtime (h)|
> |Uniform|DASBS|3.75|0.4|3.75|0.4|3.75|0.4|
> |Uniform|LEAPS|30|11.8|30|11.8|30|11.8|
> |Masked|MDNS|50|17.6|100|35.2|100|35.2|
>
> # 4. MDNS better at low temperatures
>
> We acknowledge that MDNS outperforms DASBS at low temperatures. However, this reflects a well-documented **inductive bias** difference between *uniform* and *masked* reference processes for generative modeling [1-3], rather than a flaw in our AM solver. We evaluated on uniform diffusion to cleanly isolate our solver's efficacy, where DASBS successfully achieves state-of-the-art efficiency among all uniform-based samplers.
>
> More importantly, our choice of uniform diffusion is deeply theoretically motivated. While [4] attempts discrete AM for masked diffusion, they rely on complex training iterations that completely sacrifice the methodological elegance of continuous AM. By contrast, our cyclic group structure and uniform dynamics perfectly mirror the fundamental additive noise mechanism of the original continuous AM. Our primary contribution is establishing this authentic, structurally faithful discrete counterpart to continuous AM. Adapting this level of structural elegance to the non-additive nature of masked diffusion is a non-trivial but highly exciting future direction.
>
> [1] Lou et al. Discrete Diffusion Language Modeling by Estimating the Ratios of the Data Distribution.
>
> [2] Schiff et al. Simple Guidance Mechanisms for Discrete Diffusion Models.
>
> [3] Amin et al. Why Masking Diffusion Works: Condition on the Jump Schedule for Improved Discrete Diffusion.
>
> [4] So et al. Discrete Adjoint Matching.
>
> # 5. Convergence of (ctrl-AM, corr-AM)
>
> We can prove that **(corr-DM) and (corr-AM) have the same unique fixed-point**, and thus the convergence follows by Thm. 5.2. A key step is to derive (22, 25) with $\hat\varphi\gets\hat\varphi^{(k)}$ and $p^\star\gets p^{(k)}$, using analogous arguments. We will add this result in revision.
>
> # 6. Cyclic structure for extending AM
>
> Please refer to point 2 in our response to reviewer HN4f.
>
> We thank the reviewer again for the meticulous review and are happy to provide further explanations if needed.

---

> > ### Author Rebuttal · Reviewer_1dhf · 2026-04-03
> >
> > The authors fully addressed my concerns.

---

### Decision · Program_Chairs · 2026-04-30

**Decision:**

Accept (regular)

**Comment:**

This paper makes a meaningful methodological contribution by extending adjoint matching/ASBS to discrete spaces, with a solid theoretical treatment of the discrete SB/SOC connection and accompanying convergence analysis. Reviewers broadly agreed on the paper’s novelty, technical soundness, and strong efficiency advantages. The main concerns (cyclic-group assumptions, limited synthetic evaluation, scalability, and free-energy estimation) were all addressed in the rebuttal, and the reviewers indicated that the remaining issues are not decision-critical.  I therefore recommend accept.